# Random Normalization Aggregation for Adversarial Defense

**Minjing Dong**[1], **Xinghao Chen**[2], **Yunhe Wang**[2], **Chang Xu**[1*]
[1]School of Computer Science, University of Sydney
[2]Huawei Noah's Ark Lab
mdon0736@uni.sydney.edu.au, xinghao.chen@huawei.com,
yunhe.wang@huawei.com, c.xu@sydney.edu.au

## Abstract

The vulnerability of deep neural networks has been widely found in various models as well as tasks where slight perturbations on the inputs could lead to incorrect predictions. These perturbed inputs are known as adversarial examples and one of the intriguing properties of them is *Adversarial Transfersability*, *i.e.* the capability of adversarial examples to fool other models. Traditionally, this transferability is always regarded as a critical threat to the defense against adversarial attacks, however, we argue that the network robustness can be significantly boosted by utilizing adversarial transferability from a new perspective. In this work, we first discuss the influence of different popular normalization layers on the adversarial transferability, and then provide both empirical evidence and theoretical analysis to shed light on the relationship between normalization types and transferability. Based on our theoretical analysis, we propose a simple yet effective module named Random Normalization Aggregation (RNA) which replaces the batch normalization layers in the networks and aggregates different selected normalization types to form a huge random space. Specifically, a random path is sampled during each inference procedure so that the network itself can be treated as an ensemble of a wide range of different models. Since the entire random space is designed with low adversarial transferability, it is difficult to perform effective attacks even when the network parameters are accessible. We conduct extensive experiments on various models and datasets, and demonstrate the strong superiority of proposed algorithm. The PyTorch code is available at `https://github.com/UniSerj/Random-Norm-Aggregation` and the MindSpore code is available at `https://gitee.com/mindspore/models/tree/master/research/cv/RNA`.

## 1 Introduction

Deep Neural Networks (DNNs) have achieved impressive performance in various tasks [1, 2, 3]. However, it is well known that DNNs are susceptible to maliciously generated adversarial examples [4, 5]. Through imperceptible perturbations on the model inputs during inference stage, the model is misled to wrong predictions at a high rate. Since then, a wide range of attack techniques have been proposed under different settings and show strong attack capability. For example, attackers have full access to the model architecture and parameters, which forms white-box attacks [5, 6], and attackers have limited query access to the model, which forms black-box attacks [7, 8]. Since the high attack success rates of these techniques reveal the high risk of DNNs, defenses against adversarial examples have received increasing attention and adversarial robustness becomes one of the key criteria.

---

[*]Corresponding author.

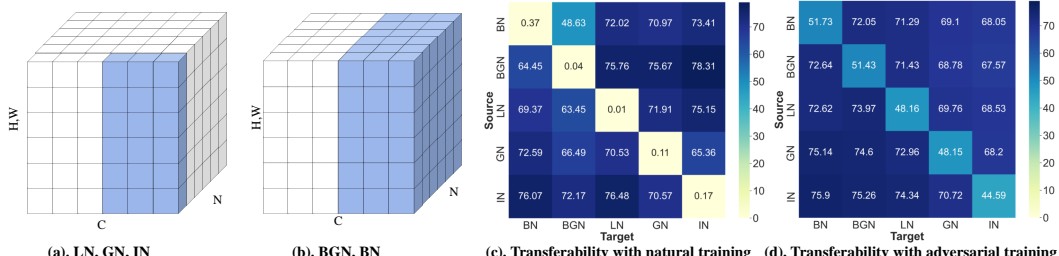

Figure 1: (a) denotes the normalized dimensions of LN, GN, and IN, while (b) denotes BGN and BN. (c) and (d) denote the adversarial transferability among different types of normalizations.

To mitigate this risk, adversarial training is proposed to yield robust models through training on generated adversarial examples [5, 6]. Besides training procedure, some regularization and image preprocessing techniques are introduced to improve adversarial robustness [9, 10]. Recent work note that the architecture and module designs could play important roles in the robustness [11, 12]. Hence, we pay more attention to the basic modules in the network which are seldom considered for improving robustness, such as normalization layers. Existing works have discussed the influence of Batch Normalization (BN) and empirically shown that BN increases adversarial vulnerability and decreases adversarial transferability [13, 14]. However, the theoretical analysis of this observation is insufficient and how to tackle this pitfall or even utilize this property to defend attacks is unexplored.

In this work, we take numerous normalizations into consideration, including Layer Normalization (LN), Group Normalization (GN), Instance Normalization (IN), Batch Normalization (BN), and Batch Group Normalization (BGN) [15, 16, 17, 18, 19], as shown in Figure 1 (a) and (b). To evaluate the influence of different normalizations on the robustness, we first conduct PGD-7 attack [6] to both natural and adversarial trained networks with different normalizations on CIFAR-10, as shown in the diagonals of Figure 1 (c) and (d). Not surprisingly, BN obtains the best robustness compared to other variants. However, we have an intriguing observation after the transferability evaluations among different normalizations. As illustrated in the heatmaps, the adversarial accuracies in most scenarios are around 70% when fed with transferred adversarial examples, while those under white-box attack are around 50%. This huge gap mainly comes from the adversarial transferability among normalizations. Motivated by this observation, we first explore the relationship between adversarial transferability and normalizations, and show that the gradient similarity and loss smoothness are the key factors of the discrepancy in transferability among different normalizations. Based on the theoretical evidence, we propose to aggregate different types of normalizations to form random space in the inference phase where the adversarial transferability can be significantly reduced. With designed random space, the inference phase naturally forms the black-box setting for those attackers who have access to model parameters due to the colossal random space. Together with the proposed black-box adversarial training framework, the adversarial robustness is substantially improved with less reduction of natural accuracy. For example, with the same adversarial training setting, the proposed algorithm improves the natural accuracy by 2.45% and the adversarial accuracy by 8.53% with ResNet-18 on CIFAR-10 under PGD[20] attack. Our contributions can be summarized as:

**1**) We provide both empirical and theoretical evidence that the upper bound of adversarial transferability is influenced by the types and parameters of normalization layers.

**2**) We propose a novel Random Normalization Aggregation (RNA) module which replaces the normalization layers to create huge random space with weak adversarial transferability. Together with a natural black-box adversarial training, RNA boosts the defense ability.

**3**) We conduct extensive experiments to demonstrate the superiority of RNA on different benchmark datasets and networks. Different variants and components are also studied.

## 2  Related Work

DNNs are vulnerable to adversarial examples and arouse lots of research interests in the attack and defense techniques [4, 5]. Expectation Over Transformation (EOT) is introduced to generating

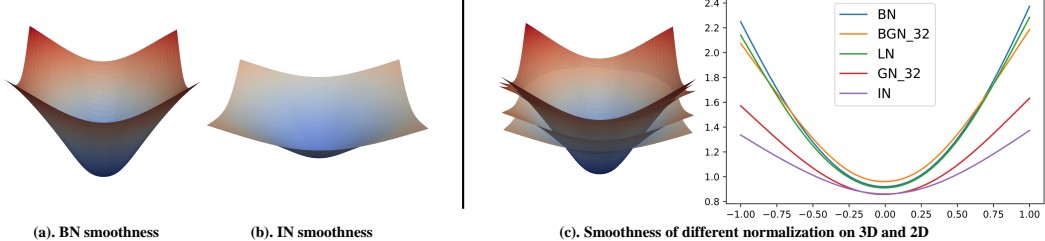

(a). BN smoothness      (b). IN smoothness      (c). Smoothness of different normalization on 3D and 2D

Figure 2: (a) and (b) denote the smoothness of loss *w.r.t.* input with BN and IN respectively. (c) denotes the smoothness of BN, BGN, LN, GN, and IN with both 3D and 2D plots.

adversarial examples by computing the gradient over the expected transformation to the input [20]. Rice *et al*. [21] explore the overfitting issue in adversarial training and propose to improve the robustness via early-stopping. Recently, the influence of DNN basic component on adversarial robustness has been paid more attention, such as activation function [22], operation [23], and neural architecture [11, 24, 25]. In terms of BN, Xie *et al*. [26] explore the robustness at different network scales and introduce a mixture of two BN layers which take care of clean and adversarial examples separately to improve the trade-offs between clean and adversarial accuracy. The mixture of BN can also improve the generalization of network with adversarial training [27]. Benz *et al*. [13] provide empirical evidence that BN increase the adversarial vulnerability. In this paper, we lay emphasis on the normalization layers and explore the connections between adversarial robustness and the aggregation of normalization layers to improve defense performance.

## 3 Adversarial Transferability with Different Normalization

In this section, we reveal the connections between adversarial transferability and normalization layers. We first consider a network which is identified with an hypothesis $h$ from a space $\mathcal{H}$. The network $h$ is optimized with the loss function $\mathcal{L}$ on input $\mathcal{X}$ and labels $\mathcal{Y}$. The objective is formulated as

$$h^* = \underset{h \in \mathcal{H}}{\operatorname{argmin}} \underset{x,y \sim \mathcal{X},\mathcal{Y}}{\mathbb{E}}[\mathcal{L}(h(x), y)]. \tag{1}$$

Given a target network $h$ and inputs $\{x, y\}$, the adversarial examples are defined as perturbed input $\tilde{x} = x + \delta$, which makes the network $h$ misclassify through maximizing the classification loss as

$$\tilde{x} = \underset{\tilde{x}:\|\tilde{x}-x\|_p \leqslant \epsilon}{\operatorname{argmax}} \mathcal{L}(h(\tilde{x}), y), \tag{2}$$

where the perturbation $\delta$ is constrained by its $l_p$-norm. Adversarial transferability denotes an inherent property of $\tilde{\mathcal{X}}$ that these adversarial examples can also boost the classification loss $\mathcal{L}(h'(\tilde{x}), y)$ of other networks as well, where $h' \in \mathcal{H}$. We empirically demonstrate that transferability is influenced by the normalization layers in the network $h$, as shown in Figure 1 (d). For example, taking BN and IN as source models to generate adversarial examples, the adversarial accuracies of LN are 71.29% and 74.34% respectively. We further provide more theoretical analysis of their relationships.

### 3.1 Definition of Normalization Layers

Batch Normalization is known as important basic module in DNNs, which improves the network performance, and a wide variety of variants are introduced where the activations are normalized with different dimensions as well as sizes. To cover most types of normalization, we divide them into two categories. An illustration is shown in Figure 1 (a) and (b), where LN, GN, and IN compute the mean and variance for each example with different group sizes during inference, while BN and BGN adopt the pre-calculated mini batch statistics which are computed by moving average in the training phase. Note that LN and IN are special cases of GN, which takes the minimum or maximum group number. Likewise, BN is a special case of BGN. For simplicity, we use GN and BGN to cover all these normalizations. Considering the activations $y \in \mathbb{R}^{d \times N}$ where $N$ denotes the batch size and $d$ denotes the number of features, the normalized outputs after BGN with group number of $s_{(BGN)}$ and

those of GN with group number of $s_{(GN)}$ during inference stage are formulated as

$$(\hat{y}_{(BGN)}^{(k)})_i = \frac{(y^{(k)})_i - (\mu_{(BGN)})_i}{(\sigma_{(BGN)})_i}, \quad (z_{(BGN)}^{(k)})_i = \gamma_{(BGN)} * (\hat{y}_{(BGN)}^{(k)})_i + \beta_{(BGN)}, \text{ for } 1 \leq k \leq N,$$

$$(\hat{y}_{(GN)}^{(k)})_i = \gamma_{(GN)} \frac{(y^{(k)})_i - (\mu_{(GN)}^{(k)})_i}{(\sigma_{(GN)}^{(k)})_i} + \beta_{(GN)}, \quad (z_{(GN)}^{(k)})_i = \gamma_{(GN)} * (\hat{y}_{(GN)}^{(k)})_i + \beta_{(GN)}, \text{ for } 1 \leq k \leq N, \quad (3)$$

$$\text{where } (\mu_{(GN)}^{(k)})_i = \frac{1}{G} \sum_{j=1}^{G} (y^{(k)})_{G\lfloor \frac{i}{G} \rfloor + j}, \quad (\sigma_{(GN)}^{(k)})_i = \sqrt{\frac{1}{G} \sum_{j=1}^{G} ((y^{(k)})_{G\lfloor \frac{i}{G} \rfloor + j} - (\mu_{(GN)}^{(k)})_i)^2},$$

where $G = \lfloor \frac{d}{s_{(GN)}} \rfloor$ denotes the group size of GN, $(\mu_{(BGN)})_i$ and $(\sigma_{(BGN)})_i$ denote the tracked mean and standard deviation of group $\lfloor \frac{i \cdot s_{(BGN)}}{d} \rfloor$.

## 3.2 Variation of Loss Function Smoothness

Existing work on adversarial transferability reveals that the adversarial transferability is mainly influenced by the dimensionality of the space of adversarial examples, since the adversarial subspaces of two networks are more likely to intersect with the growth of this dimensionality. [5, 28]. The size of space of adversarial examples can be estimated by the maximum number of orthogonal vectors $r_i$ which are aligned with the gradient $g = \nabla_{\mathcal{X}} \mathcal{L}(h(\mathcal{X}), \mathcal{Y})$. In [28], a tight bound is derived as $g^\top r_i \geq \frac{\epsilon \|g\|_2}{\sqrt{k}}$ where $k$ denotes the maximum number of $r_i$, which implies that the smoothness of loss function is inversely proportional to the adversarial transferability. Thus, we analyze the influence of different normalization layers on the smoothness of loss function, including GN and BGN.

For simplicity, we dismiss the usage of $k$ in the following equations since the computation of both GN and BGN during inference is independent of batch size. We denote the loss with GN as $\hat{\mathcal{L}}_{gn}$ and the loss with BGN as $\hat{\mathcal{L}}_{bgn}$. Since the mean and variance are computed based on current group for both GN and BGN, we compute the partial derivative of loss w.r.t. a group $Y_j$ instead of $y_i$ where $Y_j = y_{[G\lfloor \frac{i}{G} \rfloor : G\lfloor \frac{i}{G} \rfloor + G]}$. Similarly, $Z_j$ denotes the activations of a group after normalization layers. Based on Eq. 3, the partial derivative of $\hat{\mathcal{L}}_{gn}$ and $\hat{\mathcal{L}}_{bgn}$ w.r.t. $Y_j$ can be given as

$$\frac{\partial \hat{\mathcal{L}}_{gn}}{\partial Y_j} = \frac{\gamma_{gn}}{G \cdot \sigma_j^{gn}} (G \cdot \frac{\partial \hat{\mathcal{L}}_{gn}}{\partial Z_j} - 1\langle 1, \frac{\partial \hat{\mathcal{L}}_{gn}}{\partial Z_j} \rangle - \hat{Y}_j \langle \frac{\partial \hat{\mathcal{L}}_{gn}}{\partial Z_j}, \hat{Y}_j \rangle),$$

$$\frac{\partial \hat{\mathcal{L}}_{bgn}}{\partial Y_j} = \frac{\gamma_{bgn}}{\sigma_j^{bgn}} \frac{\partial \hat{\mathcal{L}}_{bgn}}{\partial Z_j}, \quad (4)$$

where $\langle, \rangle$ denotes the inner product, $\sigma_j^{gn}$ denotes the standard deviation of $Y_j$, and $\sigma_j^{bgn}$ denotes the tracked standard deviation of $Y_j$. For simplicity, we denote $\hat{g} = \frac{\partial \hat{\mathcal{L}}}{\partial Y_j}$ and $g = \frac{\partial \hat{\mathcal{L}}}{\partial Z_j}$. Taking the advantage of the fact that the mean of $Y_j$ is zero and its norm is $\sqrt{G}$, the squared norm of the partial derivative of GN and BGN can be derived as

$$\|\hat{g}_{gn}\|^2 = \frac{\gamma_{gn}^2}{(\sigma_j^{gn})^2} (\|g_{gn}\|^2 - \frac{1}{G}\langle 1, g_{gn} \rangle^2 - \frac{1}{G}\langle g_{gn}, \hat{Y}_j \rangle^2), \quad \|\hat{g}_{bgn}\|^2 = \frac{\gamma_{bgn}^2}{(\sigma_j^{bgn})^2} \|g_{bgn}\|^2. \quad (5)$$

Besides the smoothness of the loss, we further consider the smoothness of the gradients of the loss for GN and BGN. Following [29], we compute the "effective" $\beta$-smoothness through the quadratic form of Hessian of the loss w.r.t. the group activations in the normalized gradient direction, which measures the change of gradients with perturbations in the gradient direction. For simplicity, we denote the hessian w.r.t. the layer output as $\hat{H} = \frac{\partial \hat{\mathcal{L}}}{\partial Y_j \partial Y_j}$, the hessian w.r.t. the normalization output as $H = \frac{\partial \hat{\mathcal{L}}}{\partial Z_j \partial Z_j}$, the normalized gradient as $\hat{g}' = \frac{\hat{g}}{\|\hat{g}\|}$ and $g' = \frac{g}{\|g\|}$. For GN and BGN, we have

$$\hat{g}_{gn}'^\top \hat{H}_{gn} \hat{g}_{gn}' \leq \frac{\gamma_{gn}^2}{(\sigma_j^{gn})^2} \left[ g_{gn}'^\top H_{gn} g_{gn}' - \frac{1}{G \cdot \gamma_{gn}} \langle g_{gn}, \hat{Y}_j \rangle \right], \quad \hat{g}_{bgn}'^\top \hat{H}_{bgn} \hat{g}_{bgn}' \leq \frac{\gamma_{bgn}^2}{(\sigma_j^{bgn})^2} \left[ g_{bgn}'^\top H_{bgn} g_{bgn}' \right]. \quad (6)$$

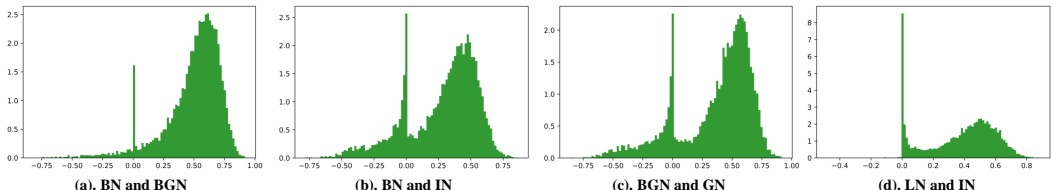

(a). BN and BGN      (b). BN and IN      (c). BGN and GN      (d). LN and IN

Figure 3: The histograms over the gradient cosine similarity of different normalization layers.

### 3.3 Normalization Layers and Adversarial Transferability

The sufficient conditions and the bounds of adversarial transferability between two networks have been discussed in [30]. We extend this result to the networks with different normalization layers. Since we focus on the influence of different normalization layers, we assume that these networks share the same loss function and weight parameters $W$, which makes $\frac{\partial \hat{\mathcal{L}}_{gn}}{\partial Z_j} = \frac{\partial \hat{\mathcal{L}}_{bgn}}{\partial Z_j}$ and $\frac{\partial \hat{\mathcal{L}}_{gn}}{\partial Z_j \partial Z_j} = \frac{\partial \hat{\mathcal{L}}_{bgn}}{\partial Z_j \partial Z_j}$.
Meanwhile, Eq. 5 and Eq. 6 can be easily generalized to the input $x$ since $\frac{\partial \hat{\mathcal{L}}}{\partial x} = \frac{\partial \hat{\mathcal{L}}}{\partial Y} W$. With this assumption, the connections between normalization layers and adversarial transferability can be established via bounded gradient norm and $\beta$-smoothness in Eq. 5 and Eq. 6 as

**Theorem 3.1.** *Given two networks $h_a$ and $h_b$ with different normalization layers, the adversarial perturbation under white-box attack is $\delta$ on $x$ with attack target label $y_{\mathcal{A}}$ and true label $y_{\mathcal{T}}$. Assume $h_a$ and $h_b$ are "effective" $\beta_a$ and $\beta_b$-smooth respectively, the level of adversarial transferability $T$ between networks $h_a$ and $h_b$ within the perturbation ball $\|\delta\|_2 \leq \epsilon$ can be upper bounded by*

$$T \leq \frac{\mathcal{R}_a + \mathcal{R}_b}{\min(\mathcal{L}(x, y_{\mathcal{A}})) - \max(\|\nabla_x \mathcal{L}\|)\epsilon(\sqrt{\frac{1+\bar{S}}{2}} + 1) - \max(\beta_a, \beta_b)\epsilon^2}, \tag{7}$$

*where $T$ denotes the attack successful rate, $\mathcal{R}_a$ and $\mathcal{R}_b$ denotes the empirical risks of network $h_a$ and $h_b$, $\bar{S}$ denotes the upper loss gradient similarity, $\min(\mathcal{L}(x, y_{\mathcal{A}})) = \min_{x \sim \mathcal{X}}(\mathcal{L}_a(x, y_{\mathcal{A}}), \mathcal{L}_b(x, y'))$, and $\max(\|\nabla_x \mathcal{L}\|) = \max_{x \sim \mathcal{X}, y \sim \{y_{\mathcal{T}}, y_{\mathcal{A}}\}}(\|\nabla_x \mathcal{L}_a(x, y)\|, \|\nabla_x \mathcal{L}_b(x, y)\|)$. Since the networks share the same loss function and weight parameters, we denote the influence of weight parameters as some constant $C_g$ on gradient norm and $C_H$ on gradient smoothness. The partial derivative and Hessian of loss w.r.t. the normalization output are the same for different normalization, denoted as $g$ and $H$ respectively. The gradient norm, $\beta_a$, and $\beta_b$ in Eq. 7 can be bounded as*

$$\|\nabla_x \mathcal{L}\| \leq C_g \cdot \max\left(\frac{|\gamma_{gn}|}{\sigma_j^{gn}}\sqrt{\|g\|^2 - \frac{1}{G}\langle 1, g\rangle^2 - \frac{1}{G}\langle g, \hat{Y}_j\rangle^2}, \frac{|\gamma_{bgn}|}{\sigma_j^{bgn}}\|g\|\right),$$
$$\beta_{a,b} \leq C_H \cdot \max\left(\frac{\gamma_{gn}^2}{(\sigma_j^{gn})^2}\left[g'^\top H g' - \frac{1}{G \cdot \gamma_{gn}}\langle g, \hat{Y}_j\rangle\right], \frac{\gamma_{bgn}^2}{(\sigma_j^{bgn})^2}\left[g'^\top H g'\right]\right). \tag{8}$$

Combining Eq. 7 and 8, we observe that the upper bound of adversarial transferability is controlled by the gradient magnitude and gradient smoothness, which is further bounded according to the type and parameters of normalization layers. Specifically, given the same $\gamma$ and $\sigma$ for GN and BGN, GN achieves a smaller gradient norm and better gradient smoothness than BGN, which decreases the upper bound of adversarial transferability. Furthermore, the group size $G$ in GN plays an important role in smoothness. With smaller $G$, the smoothness of GN increases, and thus the upper bound of adversarial transferability decreases. Similar observations can be found in empirical evidence. As shown in Figure 2, the loss landscapes of different normalization layers *w.r.t.* input are visualized, which demonstrates that different normalization layers have different smoothness. Furthermore, IN achieves the best performance in smoothness, which corresponds to the observation in Eq. 8, since IN has the minimum group size. The attack success rate is relatively low when the source model is IN, as shown in Figure 1 (c) and (d), which corresponds to the observation in Theorem 3.1 that the adversarial transferability decreases when the network is smoother.

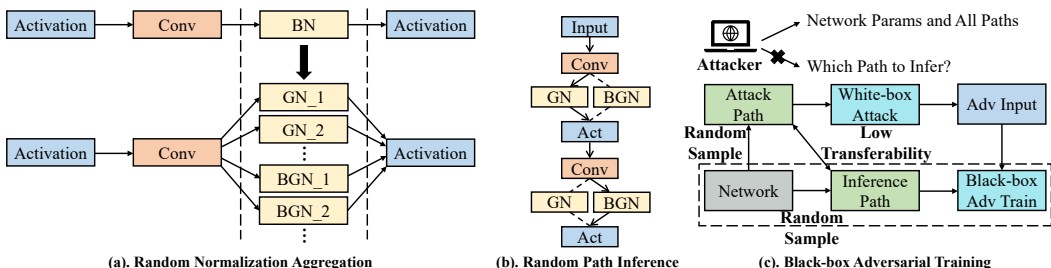

Figure 4: An illustration of Random Normalization Aggregation and Black-box Adversarial Training.

# 4  Random Normalization Aggregation

Since the adversarial transferability is strongly correlated with the type of normalization layers, we ask a simple question: *Can we utilize the bounded adversarial transferability among normalization layers to defense against white-box attacks?* In this work, we propose a Random Normalization Aggregation (RNA) module, which replaces the BN layer in the network. As shown in Figure 4 (a), the normalization layers becomes a combination of different normalization sampled from GNs and BGNs, where the underline denotes the group number. Specifically, the network maintains different normalization layers while only one normalization is randomly selected for each layer during forwarding, as shown in Figure 4 (b). Through incorporating randomization in normalization layers, the network with RNA module can be treated as a "supernet" with multiple paths. Back to white-box defense setting, we assume that the attackers have access to the network parameters. The adversarial examples are generated through backward on a randomly sampled path, and then fed to another randomly sampled path due to RNA module, which makes the entire white-box attack become a "black-box" attack, as illustrated in Figure 4 (c). Thus, together with the adversarial transferability study in Section 3, it is natural to create a network with random space in normalization layers where the adversarial transferability is significantly constrained. To achieve a strong defense against adversarial attacks, some concerns still remain: (1). The number of paths are required to be extremely large to reduce the probability of sampling the same path with random sampling strategy; (2). The collaboration with traditional adversarial training; (3). The normalization types need to be carefully selected to enforce low adversarial transferability. We discussion these concerns as follows.

**Path Increment in Random Space**   The adversarial transferability among different normalization has been discussed in Figure 1 (c) and (d). However, the size of random space also matters for effective defense against attacks. If the attackers can sample the same path during attack and inference phases with a high probability, the adversarial accuracy will decrease tremendously. To tackle this issue, we introduce layer-wise randomization of RNA module, which randomly samples the normalization for each layer in the network. As shown in Figure 4 (b), different normalization types are sampled for different layers, which exponentially increases the number of paths. Given the $n$ normalization types in RNA and $L$ layers in the network, the size of random space becomes $n^L$, which reduces the probability of sampling the same path during attack and inference phase to $\frac{1}{n^L} << \frac{1}{n}$.

**Black-box Adversarial Training**   It is natural to incorporate RNA module into adversarial training. Consistent with inference phase, we randomly sample a path $p_a$ and conduct white-box attack to generate adversarial examples $\tilde{\mathcal{X}}_{p_a}$. Different from traditional adversarial training which optimizes $p_a$ through feeding $\tilde{\mathcal{X}}_{p_a}$, we feed $\tilde{\mathcal{X}}_{p_a}$ to another randomly sampled path $p$, which forms a "black-box" adversarial training, as illustrated in Figure 4 (c). Eq. 1 and Eq. 2 can be reformulated as

$$h^* = \operatorname*{argmin}_{h \in \mathcal{H}} \ \mathbb{E}_{x,y \sim \mathcal{X},\mathcal{Y}; p \sim \mathcal{P}}[\mathcal{L}(h(\tilde{x}_{p_a}; p), y)], \ \ \text{where } \tilde{x}_{p_a} = \operatorname*{argmax}_{\tilde{x}_{p_a} : \|\tilde{x}_{p_a} - x\|_p \leqslant \epsilon} \mathcal{L}(h(\tilde{x}_{p_a}; p_a), y), \quad (9)$$

where $\mathcal{P}$ denotes the space of paths. The training procedure is shown in Algorithm 1.

**Normalization Types Selection**   In RNA module, multiple normalization types are maintained to form random space. According to Theorem 3.1, the adversarial transferability is bounded by different components, including gradient similarity, empirical risks, gradient magnitude, and gradient smoothness. We first provide empirical evidence that normalization layers from the same category defined in Section 3.1 tend to have higher gradient similarity. As shown in Figure 3, we visualize

**Algorithm 1** Random Normalization Aggregation with Black-box Adversarial Training

---

**Input:** The training tuple $\{\mathcal{X}, \mathcal{Y}\}$; Path set $\mathcal{P}$; Attack step size $\eta$; Attack iterations $t$; Perturbation size $\epsilon$; Network $h$ with parameters $W$;
Replace BN layers with RNA modules, and initialize the network.
**while** not converge **do**
    Sample a batch of data {x, y} from $\{\mathcal{X}, \mathcal{Y}\}$;
    Randomly sample a path $p_a$ from $\mathcal{P}$;
    Initialize adversarial perturbation $\delta$;
    **for** $i \leftarrow 1$ to $t$ **do**
        $\delta = clip_\epsilon[\delta + \eta \cdot \text{sign}(\nabla_x \mathcal{L}(h(x; p_a), y)];$
    **end for**
    Randomly sample a path $p$ from $\mathcal{P}$;
    $W = W - \nabla_W \mathcal{L}(h(x + \delta; p), y);$
**end while**

---

the histograms over the cosine similarity of two networks with different normalization layers. For example, BN and BGN belong to the same category, and their gradient similarity is much higher than that between BN and IN, comparing Figure 3 (a) and (b). Since the gradient similarity is proportional to the upper bounds of adversarial transferability in Eq. 7, we propose to select normalization from different categories. Thus, RNA module samples the normalization types from both GN and BGN with small group sizes in our experiments, while the evaluation of other combinations is also included.

## 5 Experiments

In this section, we provide sufficient evaluation of RNA module on various models and datasets.

### 5.1 Evaluation Setup

**CIFAR-10/100** We first conduct experiments on CIFAR-10/100 [31] datasets, which contain 50K training images and 10K testing images with size of 32×32 from 10/100 categories. The networks we use are ResNet-18 [31] and WideResNet-32 (WRN) [32]. The SGD optimizer with a momentum of 0.9 is used. The weight decay is set to $5 \times 10^{-4}$. The initial learning rate is set to 0.1 with a piecewise decay learning rate scheduler. All the baselines are trained with 200 epochs with a batch size of 128. The PGD-10 with $\epsilon = 8/255$ and step size of $2/255$ is adopted in the adversarial training setting. For the RNA module, we utilize BN and IN to form the random space in normalization layers. The experiments are performed on one V100 GPU using Pytorch [33] and Mindspore [34].

**ImageNet** The effectiveness of proposed RNA is also evaluated on ImageNet [35], which contains 1.2M training images and 50K testing images with size of $224 \times 224$ from 1000 categories. The networks we use are ResNet-50 [31]. The SGD optimizer with a momentum of 0.9 is used. The weight decay is set to $1 \times 10^{-4}$. The initial learning rate is set to 0.02 with a cosine learning rate scheduler. We load a pretrained ResNet-50 and then adversarailly train the network for 60 epochs with a batch size of 512. The PGD-2 with $\epsilon = 4/255$ is adopted in the adversarial training setting. For the RNA module, we utilize BGNs and GNs with group size of 1 and 2 to form the random space. The experiments are performed on eight V100 GPUs.

**Baselines and Attacks** Our proposed RNA modules replace the normalization layers in the network. Thus, various normalization layers are involved for comparison, including BN, IN, LN, GN, and BGN [15, 16, 17, 18, 19]. On CIFAR-10/100, we evaluate the robustness of all the baselines under different strong attacks from TorchAttacks [36]. For Fast Gradient Sign Method (FGSM) [4], the perturbation size $\epsilon$ is set to $8/255$. For Projected Gradient Descent (PGD) [6], $\epsilon$ is set to $8/255$ with step size of $2/255$, and the steps are set to 20. For CW attack [37], the steps are set to 1000 with learning rate of 0.01. For Momentum variant of Iterative Fast Gradient Sign Method (MIFGSM) [38], $\epsilon$ is set to $8/255$ with a step size of $2/255$, and the steps are set to 5 with decay of 1.0. For DeepFool [39], the steps are set to 50 with overshoot of 0.02. For Auto Attack [40], $\epsilon$ is set to $8/255$. On ImageNet, we evaluate the robustness of under PGD attacks with $\epsilon$ of $4/255$ and steps of 50.

Table 1: The adversarial robustness evaluation of adversarial-trained networks on CIFAR-10.

| Model | Method | Natural | FGSM | PGD$^{20}$ | CW | MIFGSM | DeepFool | AutoAttack |
|---|---|---|---|---|---|---|---|---|
| ResNet-18 | BN [18] | 81.84 | 56.70 | 52.16 | 78.46 | 54.96 | 0.35 | 47.69 |
| | BGN32 [19] | 77.28 | 54.60 | 50.71 | 73.57 | 53.05 | 0.39 | 46.12 |
| | IN [17] | 77.07 | 50.07 | 42.76 | 72.51 | 47.63 | 2.27 | 38.30 |
| | GN32 [16] | 76.69 | 52.60 | 45.66 | 72.92 | 50.26 | 0.72 | 41.83 |
| | LN [15] | 79.81 | 53.88 | 45.44 | 75.51 | 50.72 | 1.13 | 41.48 |
| | RNA(Ours) | **84.29** | **63.10** | **60.69** | **84.45** | **60.70** | **76.73** | **65.61** |
| WideResNet32 | BN [18] | 85.27 | 60.65 | 55.06 | 82.24 | 58.47 | 0.40 | 52.24 |
| | BGN32 [19] | 83.70 | 59.66 | 54.96 | 80.25 | 57.85 | 0.37 | 51.38 |
| | IN [17] | 84.11 | 58.14 | 50.37 | 80.13 | 55.37 | 1.62 | 46.81 |
| | GN32 [16] | 83.45 | 58.95 | 51.94 | 79.55 | 56.70 | 1.37 | 47.99 |
| | LN [15] | 83.24 | 57.80 | 49.74 | 79.39 | 54.68 | 0.95 | 46.44 |
| | RNA(Ours) | **86.46** | **65.73** | **63.34** | **85.68** | **62.84** | **78.18** | **67.88** |

Table 2: The adversarial robustness evaluation of adversarial-trained networks on CIFAR-100.

| Model | Method | Natural | FGSM | PGD$^{20}$ | CW | MIFGSM | DeepFool | AutoAttack |
|---|---|---|---|---|---|---|---|---|
| ResNet-18 | BN [18] | 55.81 | 31.33 | 28.71 | 50.94 | 30.26 | 0.79 | 24.48 |
| | BGN32 [19] | 53.16 | 30.11 | 27.75 | 48.74 | 29.11 | 0.54 | 23.29 |
| | IN [17] | 52.92 | 27.56 | 23.16 | 47.70 | 25.91 | 2.86 | 19.33 |
| | GN32 [16] | 51.00 | 28.32 | 25.10 | 45.82 | 27.10 | 0.79 | 21.00 |
| | LN [15] | 48.82 | 27.05 | 23.83 | 44.07 | 26.93 | 0.80 | 19.97 |
| | RNA(Ours) | **56.79** | **36.76** | **35.55** | **56.86** | **34.00** | **51.77** | **42.12** |
| WideResNet32 | BN [18] | 60.11 | 35.40 | 31.69 | 57.11 | 34.14 | 0.23 | 28.36 |
| | BGN32 [19] | 58.54 | 34.13 | 30.97 | 54.08 | 33.01 | 0.46 | 26.92 |
| | IN [17] | 56.71 | 31.96 | 28.09 | 51.98 | 30.33 | 1.83 | 24.25 |
| | GN32 [16] | 59.08 | 33.56 | 29.94 | 53.89 | 32.30 | 1.12 | 25.78 |
| | LN [15] | 57.09 | 32.92 | 29.75 | 52.19 | 31.73 | 0.79 | 25.71 |
| | RNA(Ours) | **60.57** | **37.87** | **36.04** | **60.21** | **35.58** | **55.16** | **42.43** |

## 5.2 Results for Robustness

**Main Results** We first evaluate the performance of RNA on CIFAR-10 and CIFAR-100 under different types of attacks. The detailed results are shown in Table 1 and 2. Popular normalization layers show similar performance on robustness. However, with a random space of different normalization layers, the robustness is significantly improved. Comparing RNA with other baselines, RNA consistently achieves the best performance under all attacks, and show strong superiority. For example, RNA with ResNet-18 achieves 65.61% under Auto Attack on CIFAR-10, which is 17.92% higher than BN. Similarly, RNA with WRN achieves 55.16% under DeepFool attacks on CIFAR-100, which is 53.33% higher than IN. The boosted adversarial accuracy shows strong empirical evidence that the constrained adversarial transferability in random space can provide satisfactory defense capability. Furthermore, with proposed black-box adversarial training, RNA achieves better natural accuracy. For example, RNA with WRN achieves 86.46% on CIFAR-10, which improves BN by 1.19%. We mainly attribute this improvement to the fact that the generated adversarial examples can be treated as a "weaker" attack example to other paths during optimization, which naturally achieves better trade-offs between natural and adversarial accuracy.

**Stronger PGD Attacks** We further evaluate the defense capability of RNA under stronger PGD attacks, which enhance the number of iterations and enlarge the perturbation size. The comparison with other baselines are shown in Figure 5 (a) and (b). Comparing RNA with other baselines, RNA achieves stable robustness under different attack iterations, such as 60.70% on PGD$^{10}$ and 59.94% on PGD$^{100}$. Meanwhile, the PGD accuracy of RNA is much higher than all other baselines. For example, RNA improves BN baselines by a margin of 7.93%. In terms of larger perturbation size, RNA achieves the best robustness in all the scenarios, and the lowest decrement among all the methods. Specifically, RNA achieves 80.06% with $\epsilon$ of 2/255 and 24.90% with $\epsilon$ of 20/255, whose gap is 55.16%. For comparison, the gap of BN is 64.68% and GN is 58.40%.

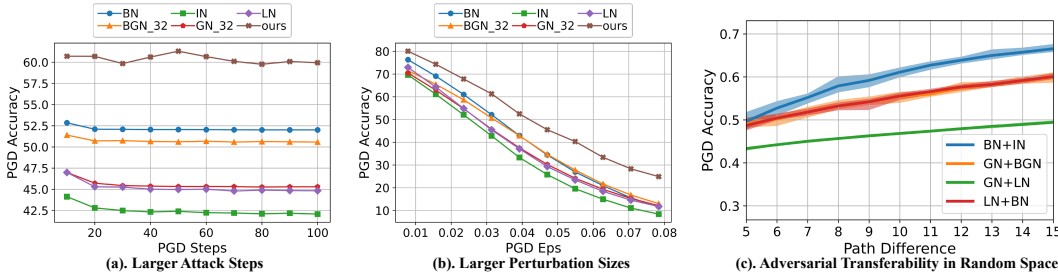

**(a). Larger Attack Steps**     **(b). Larger Perturbation Sizes**     **(c). Adversarial Transferability in Random Space**

Figure 5: The evaluation of robustness under PGD attacks settings. (a) denotes larger attack iterations, and (b) denotes larger perturbation sizes. The adversarial transferability in random space is evaluated through sampling paths with different levels of diversity in (c).

Table 3: Comparison with defense algorithms.

| Method | CIFAR-10 | | ImageNet |
| --- | --- | --- | --- |
| | $PGD^{20}$ | AA | $PGD^{50}$ |
| RobustWRN [41] | 59.13 | 52.48 | 31.14 |
| AWP [42] | 58.14 | 54.04 | - |
| RobNet [11] | 52.74 | - | 37.15 |
| RPI+RPT [43] | 53.96 | 53.30 | 42.72 |
| SAT [22] | 56.01 | 51.83 | 42.30 |
| RNA(Ours) | **63.34** | **67.88** | **54.61** |

Table 4: Robustness evaluation of random space built from different normalization combinations under different attacks.

| Normalization | $PGD^{20}$ | DeepFool | AutoAttack |
| --- | --- | --- | --- |
| BN | 52.16 | 0.35 | 47.69 |
| GN+BGN | 55.40 | 70.26 | 58.90 |
| GN+LN | 46.67 | 62.85 | 47.96 |
| LN+BN | 55.67 | 68.14 | 58.73 |
| IN+BN | **60.69** | **76.73** | **65.61** |

**Comparison with SOTA Defense Methods**   To demonstrate the superiority of RNA, we include several SOTA defense algorithms for comparison. RobustWRN [41] explores the importance of network width and depth on robustness. AWP [42] proposes to regularize the flatness of weight loss landscape to achieve robustness. RobNet [11] introduce a NAS framework for robustness. RPI+RPT [43] utilizes randomized precision for adversarial defense. SAT [44] proposes to replace ReLU with its smooth approximations, which exhibits robustness. The results are shown in Table 3. We use WRN on CIFAR-10 and ResNet-50 on ImageNet. All the baselines are evaluated under the attack of $PGD^{20}$ and AutoAttack (AA) on CIFAR-10 and $PGD^{50}$ on ImageNet. Our proposed RNA module achieves the best performance in all the scenarios. On CIFAR-10, RNA achieves $67.88\%$ under AutoAttack, with $13.84\%$ improvement compared with AWP. On ImageNet, RNA achieves $54.61\%$ under $PGD^{50}$, with $12.31\%$ improvement compared with SAT. Note that RNA replaces the normalization layer, which is orthogonal to other defense techniques. Similarly, the adversarial training in our setting can be replaced by other advanced training strategies to achieve potentially better performance.

**Adversarial Transferability in Random Space**   For a better illustration of the adversarial transferability in the random space built from RNA, we conduct the adversarial transferability study of ResNet-18 applied with RNA on CIFAR-10. We first define the path difference as the number of different normalization layers between attack and inference paths. For example, a path difference of 7 denotes two paths selecting different normalization layers in 7 layers during forwarding. For each path difference, we then randomly sample 10 path pairs for transferability evaluation. We also include different normalization combinations in RNA for comparison. The results are shown in Figure 5 (c), which illustrates the PGD accuracy under transferred attacks between path pairs along increasing path differences. The filled areas denote the maximum and minimum PGD accuracy. It is obvious that the combination of BN and IN achieves the best performance, which also corresponds to the analysis in Theorem 3.1. With random sampling strategy, the path difference is always around 10 since the number of normalization layers is 20 in this network. For example, the combination of BN and IN achieves an average PGD accuracy of $61.09\%$ when the path difference becomes 10. Compared with the baseline BN which achieves $52.16\%$ PGD accuracy, this lower adversarial transferability in our random space brings a strong defense capability against adversarial attacks.

## 5.3 Ablation Study

**Different Normalization Combinations**   We first provide more quantitative results of different combinations of normalization layers in RNA module. We conduct comparison with ResNet-18 on

Table 5: Ablation studies of RNA with ResNet-18 on CIFAR-10, including random space designing, adversarial training, and layer-based constraint.

| Random Space | ADV Train | Layer-free | Natural | FGSM | PGD[20] | CW | MIFGSM | DeepFool | AutoAttack |
|---|---|---|---|---|---|---|---|---|---|
| BN+IN+GN+LN | WhiteBox | ✗ | 10.45 | - | - | - | - | - | - |
| GN[1-64] | WhiteBox | ✗ | 75.50 | 53.62 | 49.57 | 74.63 | 51.81 | 47.20 | 50.57 |
| GN[1-64] | BlackBox | ✗ | 77.54 | 55.06 | 51.32 | 76.55 | 52.44 | 58.72 | 52.59 |
| BGN+GN[1-64] | BlackBox | ✗ | 73.81 | 53.98 | 52.98 | 73.19 | 53.07 | 63.78 | 56.86 |
| BGN+GN[1-64] | BlackBox | ✓ | 61.72 | 46.73 | 46.42 | 61.89 | 45.69 | 56.82 | 51.26 |
| BGN+GN[1-16] | BlackBox | ✓ | 70.99 | 51.66 | 49.74 | 70.48 | 51.58 | 63.63 | 55.98 |
| BGN+GN[1-4] | BlackBox | ✓ | 78.26 | 58.91 | 56.29 | 77.74 | 56.31 | 70.83 | 61.08 |
| BGN+GN[1-1] | BlackBox | ✓ | **84.29** | **63.10** | **60.69** | **84.45** | **60.70** | **76.73** | **65.61** |

CIFAR-10. As shown in Table 4, we evaluate the performance under PGD[20], DeepFool and Auto Attack, and the combination of IN and BN achieves the best robustness in all the scenarios. Consistent with the observation in Theorem 3.1, the combination of LN and BN has slightly worse performance than that of IN and BN, since IN is a smoother normalization than LN. Similarly, the combination of GN and LN achieves the worst performance, since LN is a special case of GN so that they have high gradient similarity in our empirical observation, as discussed in Figure 3. Thus, utilizing different normalization layers with smaller group size from GN and BGN, RNA module can form random space with low adversarial transferability to better improve the defense ability.

**Effectiveness of Different Components** We next demonstrate the effectiveness of each component introduced in RNA module. The detailed results are shown in Table 5 where [.] denotes the range of group size. Besides the normalization combinations, we include more discussion of the random space designing. To form a random space in normalization layers, it is natural to consider a combination of BN, IN, GN, and LN, however, it is difficult to optimize, as shown in the first row of Table 5. With the normalization layers selected from GNs, the optimization becomes stable, however, the robustness is not competitive, as shown in the second row. The involvement of black-box adversarial training significantly improves the robustness, as shown in the third row. Through expanding the random space with BGNs, the defense capability is slightly improved due to the doubled number of paths. However, the size of random space is still limited. After removing the layer-based constraint, the number of paths exponentially increases, the size of random space for each layer can be reduced for better trade-offs, as shown in the last 4 rows. Comparing BGN+GN[1-1] with GN[1-64], a better random space designing with an appropriate adversarial training strategy can achieve an improvement of 15.04% under Auto Attack, which demonstrates the necessity of these components.

# 6 Conclusions

In this paper, we explore the importance of normalization layers in adversarial robustness where the transferability among different normalization layers can be utilized to boost the defense capability. A Random Normalization Aggregation (RNA) module is proposed to form random space with low adversarial transferability for defense against adversarial attacks. We provide sufficient empirical evidence and theoretical analysis to reveal the connections between adversarial transferability and normalization types, which guides the random space designing. With the involvement of black-box adversarial training strategy and the relaxation of layer-based constraint, the robustness provided by RNA module is significantly strengthened. We demonstrate the superiority of RNA module via comprehensive experiments across different network architectures, attack settings, and benchmark datasets. Our work can provide valuable insights into the network module design for robustness.

# Acknowledgments

This work was supported in part by the Australian Research Council under Project DP210101859 and the University of Sydney Research Accelerator (SOAR) Prize. The authors acknowledge the use of the National Computational Infrastructure (NCI) which is supported by the Australian Government, and accessed through the NCI Adapter Scheme and Sydney Informatics Hub HPC Allocation Scheme. We gratefully acknowledge the support of MindSpore, CANN (Compute Architecture for Neural Networks) and Ascend AI Processor used for this research.

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
