# Random Normalization Aggregation for Adversarial Defense (Supplementary Material)

**Minjing Dong[1], Xinghao Chen[2], Yunhe Wang[2], Chang Xu[1] \***
[1]School of Computer Science, University of Sydney
[2]Huawei Noah's Ark Lab
mdon0736@uni.sydney.edu.au, xinghao.chen@huawei.com,
yunhe.wang@huawei.com, c.xu@sydney.edu.au

## A Comparison with RPI+RPT

To demonstrate the necessity of reducing the adversarial transferability in random space and the superiority of proposed black-box adversarial training, we provide more detailed comparison with RPI+RPT [1]. Note that we strictly follow the same training recipe of RPI+RPT [1] in this section. We train all the models for 160 epochs with PGD-7 adversarial training, which is different from the one introduced in Section 5.1. For a fair comparison, we consider two different attack settings.

We first consider the normal attack setting where the number of attack iterations are fixed for different examples. Specifically, PGD$^{20}$ denotes that the number of steps for PGD is 20 for all the examples. We take ResNet-18 and WideResNet32 as the models for evaluation. For the results of RPI+RPT, we use the official implementation to train the ResNet-18, and we load the pretrained models from official implementation for WideResNet32. Similar to Table 1 and 2 in the paper, we consider a wide range of attacks to evaluate the robustness, including FGSM, PGD$^{20}$, CW, MIFGSM, DeepFool and AutoAttack. The detailed results are shown in Table 1 and 2. Our proposed RNA achieves better performance under different attacks with large margins compared to RPI+RPT.

Table 1: The adversarial robustness evaluation with normal attacks settings on CIFAR-10.

| Model | Method | Natural | FGSM | PGD$^{20}$ | CW | MIFGSM | DeepFool | AutoAttack |
|---|---|---|---|---|---|---|---|---|
| ResNet-18 | RPI+RPT [1] | 82.84 | 58.31 | 52.18 | 80.02 | 56.09 | 27.81 | 49.70 |
| | RNA(Ours) | **85.33** | **63.88** | **59.79** | **84.42** | **61.05** | **76.83** | **66.35** |
| WideResNet32 | RPI+RPT [1] | 81.59 | 57.95 | 53.96 | 79.05 | 56.32 | 27.61 | 53.30 |
| | RNA(Ours) | **86.22** | **64.85** | **60.49** | **85.87** | **61.21** | **57.21** | **62.91** |

Table 2: The adversarial robustness evaluation with normal attacks settings on CIFAR-100.

| Model | Method | Natural | FGSM | PGD$^{20}$ | CW | MIFGSM | DeepFool | AutoAttack |
|---|---|---|---|---|---|---|---|---|
| ResNet-18 | RPI+RPT [1] | 56.72 | 33.04 | 29.98 | 52.87 | 31.76 | 20.07 | 28.14 |
| | RNA(Ours) | **57.62** | **36.62** | **35.51** | **57.14** | **35.84** | **53.12** | **42.41** |
| WideResNet32 | RPI+RPT [1] | 60.04 | 35.78 | 32.46 | 56.74 | 34.29 | 21.74 | 31.17 |
| | RNA(Ours) | **61.36** | **37.76** | **35.98** | **60.80** | **35.63** | **54.48** | **41.23** |

Besides the normal attack setting, we also consider a relatively weak attack setting evaluated in [1] where the attackers stop the iteration of attacks when the network misclassifies the perturbed example.

---

*Corresponding author.

Thus, the number of attack iterations can be different for different examples. Specifically, PGD[20] here denotes the maximum attack iterations for all the examples while the actual attack iterations are always smaller than 20. Under this attack, the adversarial accuracy is much higher due to the adversarial transferability. The comparison under this adaptive attack setting is provided in Table 3. Similarly, we include ResNet-18 and WideResNet32 for comparison on CIFAR-10/100. Our proposed RNA achieves better performance than RPI+RPT with large margins under a relatively weak adaptive PGD[20] attack.

Table 3: The adversarial robustness evaluation under adaptive attack setting on CIFAR-10/100.

| Model | Method | CIFAR-10 | | CIFAR-100 | |
|---|---|---|---|---|---|
| | | Natural | PGD[20] | Natural | PGD[20] |
| ResNet-18 | RPI+RPT [1] | 82.64 | 65.77 | 56.97 | 41.75 |
| | RNA(Ours) | **85.33** | **78.58** | **57.62** | **51.62** |
| WideResNet32 | RPI+RPT [1] | 81.52 | 66.75 | 58.41 | 40.45 |
| | RNA(Ours) | **86.22** | **78.33** | **61.36** | **53.55** |

# B Proofs of Theorem 2.1

**Theorem B.1.** *Given two networks $h_a$ and $h_b$ with different normalization layers, the adversarial perturbation under white-box attack is $\delta$ on $x$ with attack target label $y_\mathcal{A}$ and true label $y_\mathcal{T}$. Assume $h_a$ and $h_b$ are "effective" $\beta_a$ and $\beta_b$-smooth respectively, the level of adversarial transferability $T$ between networks $h_a$ and $h_b$ within the perturbation ball $\|\delta\|_2 \leq \epsilon$ can be upper bounded by*

$$T \leq \frac{\mathcal{R}_a + \mathcal{R}_b}{\min(\mathcal{L}(x, y_\mathcal{A})) - \max(\|\nabla_x\mathcal{L}\|)\epsilon(\sqrt{\frac{1+\bar{S}}{2}} + 1) - \max(\beta_a, \beta_b)\epsilon^2}, \tag{1}$$

*where $T$ denotes the attack successful rate, $\mathcal{R}_a$ and $\mathcal{R}_b$ denotes the empirical risks of network $h_a$ and $h_b$, $\bar{S}$ denotes the upper loss gradient similarity, $\min(\mathcal{L}(x, y_\mathcal{A})) = \min_{x \sim \mathcal{X}}(\mathcal{L}_a(x, y_\mathcal{A}), \mathcal{L}_b(x, y'))$, and $\max(\|\nabla_x\mathcal{L}\|) = \max_{x \sim \mathcal{X}, y \sim \{y_\mathcal{T}, y_\mathcal{A}\}}(\|\nabla_x\mathcal{L}_a(x, y)\|, \|\nabla_x\mathcal{L}_b(x, y)\|)$. Since the networks share the same loss function and weight parameters, we denote the influence of weight parameters as some constant $C_g$ on gradient norm and $C_H$ on gradient smoothness. The partial derivative and Hessian of loss w.r.t. the normalization output are the same for different normalization, denoted as $g$ and $H$ respectively. The gradient norm and $\beta$ in Eq. 1 can be bounded as*

$$
\begin{aligned}
\|\nabla_x\mathcal{L}\| &\leq C_g \cdot \max\left(\frac{|\gamma_{gn}|}{\sigma_j^{gn}}\sqrt{\|g\|^2 - \frac{1}{G_{gn}}\langle 1, g\rangle^2 - \frac{1}{G_{gn}}\langle g, \hat{Y}_j\rangle^2}, \frac{|\gamma_{bgn}|}{\sigma_j^{bgn}}\|g\|\right), \\
\beta &\leq C_H \cdot \max\left(\frac{\gamma_{gn}^2}{(\sigma_j^{gn})^2}\left[g'^\top H g' - \frac{1}{G_{gn}\gamma_{gn}}\langle g, \hat{Y}_j\rangle\right], \frac{\gamma_{bgn}^2}{(\sigma_j^{bgn})^2}\left[g'^\top H g'\right]\right).
\end{aligned}
\tag{2}
$$

*Proof.* Following [2], we further extend the upper bound of adversarial transferability to the network with different normalization layers. To establish the connections between normalization types and adversarial transferability, we first provide some useful simple but useful facts of different normalization layers, including Group Norm (GN) and Batch Group Norm (BGN).

During the inference stage, the computation of both GN and BGN are independent of batch size. Thus, we dismiss the discussion of batch. We first consider the network with GN. Given the activations $y \in \mathbb{R}^d$ where $d$ denotes the number of features, the normalized outputs after GN with group number of $g$ and during inference stage are formulated as

$$\hat{y}_i = \gamma\frac{y_i - \mu(i)}{\sigma(i)} + \beta, \quad z_i = \gamma * \hat{y}_i + \beta,$$

$$\text{where} \quad \mu(i) = \frac{1}{\lfloor\frac{d}{g}\rfloor}\sum_{i=0}^{\lfloor\frac{d}{g}\rfloor - 1} y_{\lfloor\frac{i\cdot g}{d}\rfloor + i}, \quad \sigma(i) = \sqrt{\frac{1}{\lfloor\frac{d}{g}\rfloor}\sum_{i=0}^{\lfloor\frac{d}{g}\rfloor - 1}(y_{\lfloor\frac{i\cdot g}{d}\rfloor + i} - \mu(i))^2}, \tag{3}$$

For simplicity, we denote the group size $G = \lfloor \frac{d}{g} \rfloor$ and a group of activations $Y_j = y_{[\lfloor \frac{i \cdot g}{d} \rfloor : \lfloor \frac{i \cdot g}{d} \rfloor + G]}$.
The partial derivative of the loss $\hat{\mathcal{L}}$ w.r.t. $y$ for GN is given as

$$
\begin{aligned}
\frac{\partial \hat{\mathcal{L}}}{\partial Y_j} &= \frac{\partial \hat{\mathcal{L}}}{\partial \hat{Y}_j} \frac{\partial \hat{Y}_j}{\partial Y_j} + \frac{\partial \hat{\mathcal{L}}}{\partial \mu_j} \frac{\partial \mu_j}{\partial Y_j} + \frac{\partial \hat{\mathcal{L}}}{\partial \sigma_j} \frac{\partial \sigma_j}{\partial Y_j} \\
&= (\frac{\partial \hat{\mathcal{L}}}{\partial \hat{Y}_j} \frac{\partial \hat{Y}_j}{\partial Y_j}) + (\frac{\partial \hat{\mathcal{L}}}{\partial \hat{Y}_j} \frac{\partial \hat{Y}_j}{\partial \mu_j} + \frac{\partial \hat{\mathcal{L}}}{\partial \sigma_j} \frac{\partial \sigma_j}{\partial \mu_j}) \frac{\partial \mu_i}{\partial Y_j} + (\frac{\partial \hat{\mathcal{L}}}{\partial \hat{Y}_j} \frac{\partial \hat{Y}_j}{\partial \sigma_j}) \frac{\partial \sigma_j}{\partial Y_j} \\
&= \frac{1}{\sigma_j} \frac{\partial \hat{\mathcal{L}}}{\partial \hat{Y}_j} + \frac{1}{G} ((\sum_{i=1}^{G} \frac{-1}{\sigma_j} \frac{\partial \hat{\mathcal{L}}}{\partial \hat{y}_i}) + \frac{\partial \hat{\mathcal{L}}}{\partial \sigma_j} \frac{(\sigma_j)^{-1}}{2G} \sum_{i=1}^{G} -2(y_i - \mu_j))) + (-1 \sum_{i=1}^{G} \frac{\partial \hat{\mathcal{L}}}{\partial \hat{y}_i} (\sigma_j)^{-2} (y_i - \mu_j)) \frac{\partial \sigma_j}{\partial Y_i} \\
&= \frac{1}{\sigma_j} \frac{\partial \hat{\mathcal{L}}}{\partial \hat{Y}_j} + \frac{1}{G} (\sum_{i=1}^{G} \frac{-1}{\sigma_j} \frac{\partial \hat{\mathcal{L}}}{\partial \hat{y}_i}) + \frac{-(\sigma_j)^{-1}}{2} \frac{2(Y_j - \mu_j)}{G} (\sum_{i=1}^{G} \frac{\partial \hat{\mathcal{L}}}{\partial \hat{y}_i} (\sigma_j)^{-2} (y_i - \mu_j)) \\
&= \frac{1}{G\sigma_j} (G \frac{\partial \hat{\mathcal{L}}}{\partial \hat{Y}_j} - \sum_{i=1}^{G} \frac{\partial \hat{\mathcal{L}}}{\partial \hat{y}_i} - \frac{(Y_j - \mu_j)}{\sigma_j} \sum_{i=1}^{G} \frac{\partial \hat{\mathcal{L}}}{\partial \hat{y}_i} \frac{(y_i - \mu_j)}{\sigma_j}) \\
&= \frac{\gamma_j}{G\sigma_j} (G \frac{\partial \hat{\mathcal{L}}}{\partial Z_j} - \sum_{i=1}^{G} \frac{\partial \hat{\mathcal{L}}}{\partial z_i} - \hat{Y}_j \sum_{i=1}^{G} \frac{\partial \hat{\mathcal{L}}}{\partial z_i} \frac{(y_i - \mu_j)}{\sigma_j}).
\end{aligned}
\tag{4}
$$

Eq. 4 can be vectorized as

$$
\frac{\partial \hat{\mathcal{L}}}{\partial Y_j} = \frac{\gamma_j}{G\sigma_j} (G \frac{\partial \hat{\mathcal{L}}}{\partial Z_j} - 1 \langle 1, \frac{\partial \hat{\mathcal{L}}}{\partial Z_j} \rangle - \hat{Y}_j \langle \frac{\partial \hat{\mathcal{L}}}{\partial Z_j}, \hat{Y}_j \rangle).
\tag{5}
$$

Let $\mu_g = \frac{1}{G} \langle 1, \frac{\partial \hat{\mathcal{L}}}{\partial Z_j} \rangle$. Note that $\|\hat{Y}_j\| = \sqrt{G}$. Eq. 5 can be written as

$$
\frac{\partial \hat{\mathcal{L}}}{\partial Y_j} = \frac{\gamma_j}{\sigma_j} ((\frac{\partial \hat{\mathcal{L}}}{\partial Z_j} - 1\mu_g) - \frac{\hat{Y}_j}{\|\hat{Y}_j\|} \langle \frac{\partial \hat{\mathcal{L}}}{\partial Z_j} - 1\mu_g, \frac{\hat{Y}_j}{\|\hat{Y}_j\|} \rangle)
\tag{6}
$$

The squared norm of the partial derivative can be derived as

$$
\begin{aligned}
\|\frac{\partial \hat{\mathcal{L}}}{\partial Y_j}\|^2 &= \frac{\gamma_j^2}{(\sigma_j)^2} (\|(\frac{\partial \hat{\mathcal{L}}}{\partial z_j} - 1\mu_g)\|^2 - \langle \frac{\partial \hat{\mathcal{L}}}{\partial z_j} - 1\mu_g, \frac{\hat{y}_j}{\|\hat{y}_j\|} \rangle^2) \\
&= \frac{\gamma_j^2}{(\sigma_i)^2} (\|\frac{\partial \hat{\mathcal{L}}}{\partial Z_j}\|^2 - \frac{1}{G} \langle 1, \frac{\partial \hat{\mathcal{L}}}{\partial Z_j} \rangle^2 - \frac{1}{G} \langle \frac{\partial \hat{\mathcal{L}}}{\partial Z_j}, \hat{Y}_j \rangle^2)
\end{aligned}
\tag{7}
$$

Similarly, we consider the network with BGN. Given the activations $y \in \mathbb{R}^d$ where $d$ denotes the number of features, the normalized outputs after BGN with group number of $g$ and during inference stage are formulated as

$$
\hat{y}_i = \gamma \frac{y_i - \mu(i)}{\sigma(i)} + \beta, \quad z_i = \gamma * \hat{y}_i + \beta,
\tag{8}
$$

where $\mu(i)$ and $\sigma(i)$ here denote the tracked mean and standard deviation which are fixed during inference stage. Similarly, we denote the group size $G = \lfloor \frac{d}{g} \rfloor$ and a group of activations $Y_j = y_{[\lfloor \frac{i \cdot g}{d} \rfloor : \lfloor \frac{i \cdot g}{d} \rfloor + G]}$. The partial derivative of the loss $\hat{\mathcal{L}}$ w.r.t. $y$ for BGN is given as

$$
\frac{\partial \hat{\mathcal{L}}}{\partial Y_j} = \frac{\partial \hat{\mathcal{L}}}{\partial Z_j} \frac{\partial Z_j}{\partial \hat{Y}_j} \frac{\partial \hat{Y}_j}{\partial Y_j} = \frac{\gamma_j}{\sigma_j} \frac{\partial \hat{\mathcal{L}}}{\partial Z_j},
\tag{9}
$$

The squared norm of the partial derivative can be derived as

$$
\|\frac{\partial \hat{\mathcal{L}}}{\partial Y_j}\|^2 = \frac{\gamma_j^2}{\sigma_j^2} \|\frac{\partial \hat{\mathcal{L}}}{\partial Z_j}\|^2,
\tag{10}
$$

Note that Eq. (5) in the paper is a combination of Eq. 7 and Eq. 10. We denote the influence of weight parameters as some constant $C_g$ on gradient norm since we assume the networks are identical except for the normalization layers. And the partial derivative of the loss w.r.t. the output of normalization

$g = \frac{\partial \hat{\mathcal{L}}}{\partial Z_j}$ is identical for networks due to the same loss function and weight parameters. Combining the assumptions with Eq. 7 and Eq. 10, we can derive the upper bound of the gradient norm through $Y_j$ of networks with GN and GBN as

$$\|\nabla_x \hat{\mathcal{L}}_{gn}\| \leq C_g \cdot \frac{|\gamma_{gn}|}{\sigma_j^{gn}} \sqrt{\|g\|^2 - \frac{1}{G}\langle 1, g\rangle^2 - \frac{1}{G}\langle g, \hat{Y}_j\rangle^2},$$

$$\|\nabla_x \hat{\mathcal{L}}_{bgn}\| \leq C_g \cdot \frac{|\gamma_{bgn}|}{\sigma_j^{bgn}} \|g\|, \tag{11}$$

Following [3], we can generalize the results of loss smoothness to the gradient smoothness via Hessian. Note that the partial derivative of GN during inference in Eq. 7 has the same format of the partial derivative of BN during training which is studied in [3] as

$$\|\frac{\partial \hat{\mathcal{L}}}{\partial y_j}\|^2 = \frac{\gamma^2}{\sigma^2}(\|\frac{\partial \hat{\mathcal{L}}}{\partial z_j}\|^2 - \frac{1}{m}\langle 1, \frac{\partial \hat{\mathcal{L}}}{\partial z_j}\rangle^2 - \frac{1}{m}\langle \frac{\partial \hat{\mathcal{L}}}{\partial z_j}, \hat{y_j}\rangle^2) \tag{12}$$

The differences are that $Y_j$ denotes a group of activations in GN while $y_j$ denotes a batch of activations in BN, and $G$ denotes the size of group while $m$ denotes the batch size. Thus, we can easily generalize the smoothness of gradient in BN during training phase which is discussed in [3] to the one in GN during inference phase. In [3], the "effective" $\beta$-smoothness is defined as the changes of gradients as we move in the direction of gradients, which corresponds to $\hat{g}^T \hat{H} \hat{g}$ where $\hat{H}$ denotes the hessian *w.r.t.* the output of activations. We slightly modify $\hat{g}$ to $\hat{g}' = \frac{\hat{g}}{\|\hat{g}\|}$ so that $\hat{g}'^T \hat{H} \hat{g}'$ corresponds to the $\beta$ smoothness in the direction of gradients. Through replacing a batch of activation $y_j$ to a group of activation $Y_j$, batch size $m$ to group size $G$, and $g$ to normalized one $g'$, the upper bound of gradient smoothness of BN in [3] can be reformulated for GN as

$$\hat{g}'^T \hat{H} \hat{g}' \leq \frac{\gamma^2}{\sigma^2}\left[g'^\top H g' - \frac{1}{G\gamma}\langle g, \hat{Y}_j\rangle\right] \tag{13}$$

Since BGN uses fixed mean and variance during inference, Tthe upper bound of gradient smoothness of BGN during inference can be derived as

$$\hat{g}'^T \hat{H} \hat{g}' \leq \frac{\gamma^2}{\sigma^2}[g'^\top H g'] \tag{14}$$

Similarly, we denote the influence of weight parameters as some constant $C_H$ on gradient smoothness since we assume the networks are identical except for the normalization layers. And the partial derivative of the loss *w.r.t.* the output of normalization $H = \frac{\partial \hat{\mathcal{L}}}{\partial Z_j \partial Z_j}$ is identical for networks due to the same loss function and weight parameters. The $\beta$-smoothness of GN and BGN in the direction of gradients can be upper bounded as

$$\beta_{gn} \leq C_H \cdot \frac{\gamma_{gn}^2}{(\sigma_j^{gn})^2}\left[g'^\top H g' - \frac{1}{G\gamma_{gn}}\langle g, \hat{Y}_j\rangle\right],$$

$$\beta_{bgn} \leq C_H \cdot \frac{\gamma_{bgn}^2}{(\sigma_j^{bgn})^2}\left[g'^\top H g'\right], \tag{15}$$

With the results in Eq. 11 and Eq. 15, we can easily extend the upper bound of adversarial transferability in [2] to the networks with GN and BGN through replacing the assumption of $\|\nabla_x \mathcal{L}\| \leq B$ with the upper bound in Eq. 11 and the assumption of $\beta$-smoothness to those in Eq.15 since the usage of $\beta$-smoothness in the upper bound of adversarial transferability in [2] lies in

$$\mathcal{L}(x, y) + \delta \cdot \nabla_x \mathcal{L}(x, y) + \frac{\beta}{2}\|\delta\|^2 \geq \mathcal{L}(x + \delta, y), \tag{16}$$

where the perturbation $\delta$ is generated by adversarial attack via gradient ascent. Thus, the assumption of $\beta$-smoothness in [2] is equivalent to the $\beta$-smoothness in the direction of gradients in Eq. 15. Finally, combining the upper bound of transferability in [2] with Eq. 11 and Eq. 15 via max function gives the desired results. $\qquad\square$

## C   Stability of Robustness

During the inference phase, we randomly sample paths from the random space we build so that the adversarial robustness could vary according to the sampled paths. To evaluate the stability of robustness with RNA module, we rerun the evaluation for 10 times with ResNet-18 on CIFAR-10. The results are shown in Table 4. The maximum accuracy is 60.72% and the minimum accuracy is 59.15%. The average accuracy of 10 runs is 59.94% and the variance is 0.19. Thus, although the performance depends on the randomly sampled paths, the stability of robustness is verified in empirical evaluations since the entire random space is built with weak adversarial transferability.

Table 4: Stability evaluation of adversarial robustness with RNA.

| Method | Trial 1 | Trial 2 | Trial 3 | Trial 4 | Trial 5 | Trial 6 | Trial 7 | Trial 8 | Trial 9 | Trial 10 |
|---|---|---|---|---|---|---|---|---|---|---|
| RNA(Ours) | 59.79 | 60.10 | 59.98 | 59.97 | 60.46 | 59.15 | 59.86 | 59.66 | 59.66 | 60.72 |

## D   Limitations

In our implementation, we simply utilize the random sampling strategy so that the path difference discussed in Figure 5 (c) is around the half of total number of layers in the networks. However, some other sampling strategies which can potentially achieve better performance are not discussed in our work. For example, the robustness can be further improved with increasing path difference in Figure 5 (c), and how to maximize the difference between attack path and inference path via sampling strategy could be an interesting problem for future study.