# OpenReview forum: "Random Normalization Aggregation for Adversarial Defense"
_NeurIPS.cc/2022/Conference — NeurIPS 2022 Accept_

### Official Review · Reviewer_ioBB · 2022-06-20

**Rating:** 3
**Confidence:** 5
**Soundness:** 2 fair
**Presentation:** 3 good
**Contribution:** 2 fair

**Summary:**

Researchers explore how different normalization layers affect adversarial transferability. They provide a theoretical upper bound on the adversarial transferability between normalization layers. Then, they proposed a module named Random Normalization Aggregation (RNA). RNA replaces the normalization layers in the network, and samples normalization layers randomly at each forward pass. This can generate an exponential number of possible paths, which makes it harder for attacker to exploit.


**Questions:**

Experiments questions/concerns:

1. What if the attacker randomizes a different path (p_a) at each iteration of PGD? Wouldn't it make the attacker more diverse, and therefore generalize better? Can the authors present such experiments?

2. Can the authors explain why the results on resNet-18 are better than those on WRN-32? For example, on CIFAR-10, AA results on ResNet-18 are better by almost 4% compared to WRN-32

3. Also, for CIFAR-10, FGSM and PGD-20 acc is lower than the Auto-Attack accuracy, which doesn't make sense. Can the authors explain these results?

4. WRN-32 with BN and standard AT is essentially the standard AT method by Madry et al. [1]. It was shown that this method reaches AA accuracy of ~52-53% against AA when combined with early stopping. However, in your paper, you claim 46.44% against AA. Did the authors use early stopping for all methods? If not, I think that should present results with early stopping [2] for a fair comparison. Otherwise, the results can be linked to adversarial overfitting [2].

5. It would be interesting to see how the method works when combined with other AT methods.

6. The authors did not refer/compared to related work in the field of normalization layers and robustness [3, 4]. Can the authors compare their method to [3, 4] and present the results?

[1] Towards Deep Learning Models Resistant to Adversarial Attacks https://arxiv.org/pdf/1706.06083.pdf

[2]  Overfitting in adversarially robust deep learning http://proceedings.mlr.press/v119/rice20a

[3] INTRIGUING PROPERTIES OF ADVERSARIAL TRAINING AT SCALE https://arxiv.org/pdf/1906.03787.pdf

[4] Adversarial Examples Improve Image Recognition https://openaccess.thecvf.com/content_CVPR_2020/html/Xie_Adversarial_Examples_Improve_Image_Recognition_CVPR_2020_paper.html


miscellaneous:

1. Figure 4 has twice (b) instead of (c)

2. Line 153 - I suggest rephrasing to- "… to *defend against* white-box attacks …"

**Limitations:**

No.
Authors should discuss the limitation of adversarial training in general and their method in particular.

**Strengths And Weaknesses:**

Strengths:

1. The paper is clear and easy to follow

2. Theoretical part is sound

3. The idea to generate random normalization paths is new to the literature

Weaknesses:

1. Experiment section presents some suspicious results - the results with resNet vs. WRN are counterintuitive, and results on Auto-Attack vs. PGD also do not make sense.  It can be a sign of obfuscated gradients. Additionally, some of the reported results are not in line with known results. See the Questions section.

2. No code/models are provided for evaluation of the results. Can the authors provide the code/models/github link for additional verification?

3. No related work section. Specifically, the authors did not address some related literature that explores the same field (see Questions section).

---

> ### Author Response · Authors · 2022-08-02
> **Response to Reviewer ioBB**
>
> We thank the reviewer for the constructive comments and valuable suggestions. We follow the training setting in [a] which is suggested by the reviewer and rerun all the baselines on CIFAR-10. We find that all the baselines with WideResNet can achieve much better results. We summarize all the changes to training setting as follows:
> 1. The training epoch is set to 200 instead of 160.
> 2. The learning rate scheduler is set to piecewise decay instead of cosine.
> 3. The standard adversarial training setting is set to PGD-10 instead of PGD-7.
>
> These changes strictly follow the training setting from the official github of [a]. We believe these new experimental results will address the concerns. Currently, we are working on the experiments on CIFAR-100 and we will update the results once they are ready.
>
> [a]. Overfitting in adversarial robust deep learning
>
> ### Re Random path at each iteration of PGD.
> We generate the adversarial examples as suggested, which randomizes different paths at each iteration of PGD. We evaluate the performance of RNA under this attack with both ResNet-18 and WideResNet on CIFAR-10/100, marked as DP PGD. The attack settings including iterations, step size, and perturbation size remain the same as PGD in the paper. The results are reported as follows:
>
> | Network | dataset | DP PGD$^{20}$ |
> | ------- | ------ | --------- |
> | ResNet-18 | C-10 | 73.57 |
> | ResNet-18 | C-100 | 45.69 |
> | WideResNet | C-10 | 74.19 |
> | WideResNet | C-100 | 47.62 |
>
> After randomly sampling path in each iteration of PGD, the attack success rate becomes much lower than standard PGD. We mainly attribute it to the huge size of random space, which makes the attacker difficult to generalize to the entire space in limited number of iterations. And how to attack RNA could be an interesting topic in future research.
>
> ### Re AA results on ResNet-18 are better than WRN-32.
> We mainly attribute the lower AA results of WRN to our weak training recipe. According to the suggestions, we follow the training setting in [a] to train WRN with RNA. The results are reported in the following table:
>
> | Setting | Training | Natural | FGSM | PGD^{20} | CW | MIFGSM | DeepFool | AutoAttack |
> | ------- | ------- | ------ | ----- | --------- | ----- | ----- | -------- | ---------- |
> | ResNet-18+RNA | ours | 85.33 | 63.88 | 59.79 | 84.42 | 61.05 | 76.83 | 66.35 |
> | WRN+RNA | ours | 86.22 | 64.85 | 60.49 | 85.87 | 61.21 | 57.21 | 62.91 |
> | ResNet-18+RNA | [a] | 84.29 | 63.10 | 60.69 | 84.45 | 60.70 | 76.73 | 65.61 |
> | WRN+RNA | [a] | 86.46 | 65.73 | 63.34 | 85.68 | 62.84 | 78.18 | 67.88 |
>
> As shown in the table, both our training setting and the one of [a] achieve similar results on ResNet-18 with RNA. However, the training setting of [a] significantly improves the performance of WRN + RNA, and the adversarial accuracy of WRN is better than ResNet-18 in all the scenarios, which implies that our training setting for WRN might be relatively weak. We will revise the results in the manuscript.
>
> [a]. Overfitting in adversarial robust deep learning
>
> ### Re PGD-20 acc is lower than the Auto-Attack accuracy.
> We mainly attribute the better AA accuracy to the attacking designing of AA and our proposed defense mechanism. Note that AA generates the adversarial examples using four attacks and stopes when the adversarial examples are found. The examples remain the clean examples if none of these four techniques attacks successfully. Comparing to PGD attack which takes 20 iterations to maximize the cross-entropy loss on all examples, these remaining clean examples after AA significantly reduces the attack success rate when all the examples are fed to another path in RNA. For example, AA generates adversarial examples using four attacks for around 80% of all examples and the rest 20% remains clean examples with ResNet-18 on CIFAR-10.
>
> In white box setting, AA can generate powerful adversarial examples through multiple attack methods. However, the attack success rate of AA can be relatively low in our black box setting due to the randomness of our algorithm. RNA designs a random space in normalizations with lower adversarial transferability. In other words, an effective attack to RNA should have powerful adversarial transferability among paths with different normalizations, which cannot be achieved by AA.
>
> We also find similar results in other work [a], which involves randomness in the input images, as shown in the following table.
>
> | Method | dataset | PGD$^{20}$ | AA |
> | ------- | ------ | --------- | --- |
> | Random[b] | C-10  | 47.06     | 47.27 |
> | Hedge[b]  | C-10  | 48.31     | 51.82 |
> | RNA      | C-10  | 60.49     | 62.91 |
> | Random[b] | C-100  | 25.43     | 31.74 |
> | Hedge[b]  | C-100  | 28.23     | 35.54 |
> | RNA      | C-100  | 37.76     | 41.23 |
>
> [b]. Wu, Boxi, et al. "Attacking adversarial attacks as a defense." arXiv preprint arXiv:2106.04938 (2021).

---

> > ### Author Response · Authors · 2022-08-02
> > **Response to Reviewer ioBB (Cont.)**
> >
> > ### Re Low AA results with WRN baselines.
> > According to the suggestions, we follow the training setting in [a] to retrain all the baselines including RNA. The results are reported in the following tables.
> >
> > ResNet-18:
> >
> > | Method | Natural | FGSM | PGD$^{20}$ | CW | MIFGSM | DeepFool | AutoAttack |
> > | ------- | ------ | ----- | --------- | ----- | ----- | -------- | ---------- |
> > | BN | 81.84 | 56.70 | 52.16 | 78.46 | 54.96 | 0.35 | 47.69 |
> > | BGN32 | 77.28 | 54.60 | 50.71 | 73.57 | 53.05 | 0.39 | 46.12 |
> > | IN | 77.07 | 50.07 | 42.76 | 72.51 | 47.63 | 2.27 | 38.30 |
> > | GN32 | 76.69 | 52.60 | 45.66 | 72.92 | 50.26 | 0.72 | 41.83 |
> > | LN | 79.81 | 53.88 | 45.44 | 75.51 | 50.72 | 1.13 | 41.48 |
> > | RNA | 84.29 | 63.10 | 60.69 | 84.45 | 60.70 | 76.73 | 65.61 |
> >
> > WRN:
> >
> > | Method | Natural | FGSM | PGD$^{20}$ | CW | MIFGSM | DeepFool | AutoAttack |
> > | ------- | ------ | ----- | --------- | ----- | ----- | -------- | ---------- |
> > | BN | 85.27 | 60.65 | 55.06 | 82.24 | 58.47 | 0.40 | 52.24 |
> > | BGN32 | 83.70 | 59.66 | 54.96 | 80.25 | 57.85 | 0.37 | 51.38 |
> > | IN | 84.11 | 58.14 | 50.37 | 80.13 | 55.37 | 1.62 | 46.81 |
> > | GN32 | 83.45 | 58.95 | 51.94 | 79.55 | 56.70 | 1.37 | 47.99 |
> > | LN | 83.24 | 57.80 | 49.74 | 79.39 | 54.68 | 0.95 | 46.44 |
> > | RNA | 86.46 | 65.73 | 63.34 | 85.68 | 62.84 | 78.18 | 67.88 |
> >
> > As shown in the tables, RNA still achieves the best performance in all the scenarios after applying the training recipe in [a], which demonstrates that RNA can be easily incorporated into other defense techniques to further improve the performance.
> >
> > [a]. Overfitting in adversarial robust deep learning
> >
> > ### Re Other AT methods.
> > Following the suggestions, we replace standard AT with FAT [c] and MART [d], and we evaluate the performance of them with ResNet-18 on CIFAR-10. The results are reported as follows:
> >
> > | Method | Natural | FGSM | PGD$^{20}$ | CW | MIFGSM | DeepFool | AutoAttack |
> > | ------- | ------ | ----- | --------- | ----- | ----- | -------- | ---------- |
> > | AT | 85.33 | 63.88 | 59.79 | 84.42 | 61.05 | 76.83 | 66.35 |
> > | FAT [c] | 86.93 | 61.43 | 56.48 | 86.76 | 57.13 | 73.79 | 60.70 |
> > | MART [d] | 86.61 | 63.63 | 60.28 | 80.12 | 61.28 | 71.45 | 64.67 |
> >
> > With other AT methods, the defense performance can be different. For example, MART achieves the best PGD$^{20}$ accuracy, FAT achieves the best CW accuracy, and AT achieves the best AutoAttack accuracy. RNA can be simply incorporated into other AT methods to further improve the defense performance.
> >
> > [c]. Zhang, Jingfeng, et al. "Attacks which do not kill training make adversarial learning stronger." International conference on machine learning. PMLR, 2020.
> >
> > [d]. Wang, Yisen, et al. "Improving adversarial robustness requires revisiting misclassified examples." International Conference on Learning Representations. 2019.
> >
> > ### Re Related work.
> > Following the suggestions, we will discuss [e] and [f] in our paper. [e] proposed to use two different BN layers for clean images and adversarial images to enhance robustness and studied the role of network capacity. [f] proposed to use two different BN layers for clean images and adversarial images so that the involvement of adversarial training can improve network generalization.
> >
> > For the comparison, [e] only reported the results with deeper networks on ImageNet, such as ResNet-152, and [f] aims at improving the generalization of EfficientNet instead of the adversarial robustness. Thus, we cannot directly use the reported results in [e] and [f] for comparison. Instead, we implement the BN mixture module proposed in [e] and [f] and apply it to ResNet-18 for comparison. Note that we feed clean images to $BN_{clean}$ and adversarial examples to $BN_{adv}$ during the evaluation of BN mixture, which could be unfair for RNA. The results are reported as follows:
> >
> > | Method | Dataset | Natural | FGSM | PGD$^{20}$ | CW | MIFGSM | DeepFool | AutoAttack |
> > | ------- | ------- | ------ | ----- | --------- | ----- | ----- | -------- | ---------- |
> > | RNA | C-10 | 85.33 | 63.88 | 59.79 | 84.42 | 61.05 | 76.83 | 66.35 |
> > | bnmix[e,f] | C-10 | 95.03 | 54.60 | 44.99 | 80.87 | 51.21 | 0.89 | 42.45 |
> > | RNA | C-100 | 57.62 | 36.62 | 35.51 | 57.14 | 35.84 | 53.12 | 42.41 |
> > | bnmix[e,f] | C-100 | 72.21 | 26.41 | 21.22 | 51.92 | 24.61 | 0.67 | 19.25 |
> >
> > Although bnmix achieves better natural accuracy since bnmix trains separate BN for clean and adversarial examples, RNA achieves much better adversarial accuracy than bnmix in all the scenarios.
> >
> > [e]. Intriguing Properties if Adversarial training at Scale
> >
> > [f]. Adversarial Examples Improve Image Recognition
> >
> > ### Re Evaluation of the results.
> > We provide the evaluation code and pretrained model with an anonymous link:
> >
> > https://drive.google.com/file/d/1JZ3Q5WB7jgvGbAgVGc9BJ0Ol0nQjM2de/view
> >
> >
> > ### Re miscellaneous:
> > Thanks for the advice. We will revise the manuscript according to the suggestions.

---

> > > ### Author Response · Authors · 2022-08-07
> > > **Response to Reviewer ioBB (Cont.)**
> > >
> > > ### Re Low AA results with WRN baselines. [Updated]
> > >
> > > According to the suggestions, we also follow the training setting in [a] to retrain all the baselines including RNA on CIFAR-100. The results are reported in the following tables.
> > >
> > > ResNet-18:
> > >
> > > | Method | Natural | FGSM | PGD$^{20}$ | CW | MIFGSM | DeepFool | AutoAttack |
> > > | ------- | ------ | ----- | --------- | ----- | ----- | -------- | ---------- |
> > > | BN | 55.81 | 31.33 | 28.71 | 50.94 | 30.26 | 0.79 | 24.48 |
> > > | BGN32 | 53.16 | 30.11 | 27.75 | 48.74 | 29.11 | 0.54 | 23.29 |
> > > | IN | 52.92 | 27.56 | 23.16 | 47.70 | 25.91 | 2.86 | 19.33 |
> > > | GN32 | 51.00 | 28.32 | 25.10 | 45.82 | 27.10 | 0.79 | 21.00 |
> > > | LN | 48.82 | 27.05 | 23.83 | 44.07 | 26.93 | 0.80 | 19.97 |
> > > | RNA | 56.79 | 36.76 | 35.55 | 56.86 | 34.00 | 51.77 | 42.12 |
> > >
> > > WRN:
> > >
> > > | Method | Natural | FGSM | PGD$^{20}$ | CW | MIFGSM | DeepFool | AutoAttack |
> > > | ------- | ------ | ----- | --------- | ----- | ----- | -------- | ---------- |
> > > | BN | 60.11 | 35.40 | 31.69 | 57.11 | 34.14 | 0.23 | 28.36 |
> > > | BGN32 | 58.54 | 34.13 | 30.97 | 54.08 | 33.01 | 0.46 | 26.92 |
> > > | IN | 56.71 | 31.96 | 28.09 | 51.98 | 30.33 | 1.83 | 24.25 |
> > > | GN32 | 59.08 | 33.56 | 29.94 | 53.89 | 32.30 | 1.12 | 25.78 |
> > > | LN | 57.09 | 32.92 | 29.75 | 52.19 | 31.73 | 0.79 | 25.71 |
> > > | RNA | 60.57 | 37.87 | 36.04 | 60.21 | 35.58 | 55.16 | 42.43 |
> > >
> > >
> > > Again, RNA achieves the best performance and WRN results are better than ResNet-18 on CIFAR-100.

---

> > > > ### Comment · Reviewer_ioBB · 2022-08-07
> > > > **Response**
> > > >
> > > > I deeply appreciate the authors' efforts to answer my questions.
> > > >
> > > > However, I still have some concerns:
> > > >
> > > > 1. I'm still missing related work section in the current uploaded version. Can the authors update their version with the promised related work section?
> > > >
> > > > 2. I appreciate the new experiments the authors did with the updated training settings, however, I'm worried that the Neurips rebuttal time is not enough to update the entire manuscript in this revision.
> > > >
> > > > 3. My main concern is the Auto-Attack results: Auto-Attack consists of A-PGD (which runs first by default), which is essentially PGD with adaptive learning rate. Therefore, it makes no sense to me that auto-attack robustness will be higher than PGD robustness or FGSM robustness. So, I don't think that the authors' explanation can facilitate this significant issue. Can the authors add further explanation? why A-PGD performs worse that PGD? usually A-PGD will outperform PGD, so AA results still should be lower than PGD results.

---

> > > > > ### Author Response · Authors · 2022-08-08
> > > > > **Response to Reviewer ioBB (Cont.)**
> > > > >
> > > > > Thank you for your reply. We have uploaded a revised version of manuscript. The revised parts are highlighted with blue color.
> > > > >
> > > > > ### Re Missing related work section
> > > > > We add a related work section in the current uploaded version, which includes the discussion of mentioned reference by all the reviewers.
> > > > >
> > > > > ### Re Updating the entire manuscript
> > > > > Currently, we have updated the Table 1, 2 ,3 and Figure 5 in the experiment section, which are our major results. The remaining experiments only include Table 4 and Table 5, which contain the comparisons of our own algorithm with ResNet-18 on CIFAR-10, which will be probably completed by the end of rebuttal time. Thanks for your patience and support.
> > > > >
> > > > > ### Re Auto-Attack results
> > > > >
> > > > > We agree that Auto-Attack is a much stronger attack than PGD **in white-box attack setting**. However, this advantage is not effective in our setting since Auto-Attack is not powerful in **adversarial transferability**. Note that our proposed RNA designs a random space with lower adversarial transferability via the aggregation of normalization layers. In other words, it naturally forms a **transferred-based black-box attack setting** where the generated adversarial examples are fed to another unknown path due to random sampling strategy. Although Auto-Attack can easily generate successfully adversarial examples for a particular path of RNA, the Auto-Attack results in Table 1 and Table 2 are determined by the performance of generated adversarial examples on another randomly sampled path of RNA, which refers to the adversarial transferability of Auto-Attack.
> > > > >
> > > > > To the best of our knowledge, the adversarial transferability of Auto-Attack can be worse than PGD. For example, some empirical evidence can be found in Table 7 of [g]. We put part of the results here for illustration. The following table denotes the attack success rate of adversarial examples generated by PGD and Auto-Attack using different source models including Deit-T and Deit-B. The target models correspond to the columns, including Res152, WRN, DN201, T2T-24, T2t-7, TnT, and Vit-S.
> > > > >
> > > > > | Model | Attack | Res152 | WRN | DN201| T2T-24 | T2T-7 | TnT | ViT-S |
> > > > > | ------ | ----- |------- | ---- | ------ | ------ | ----- | --- | ----- |
> > > > > | Deit-T | PGD | 15.2 | 17.9 | 20.3 | 9.6 | 30.8 | 33.6 | 63.7 |
> > > > > | Deit-T | AA | 9.0 | 8.9 | 9.3 | 4.8 | 18.8 | 10.3 | 33.7 |
> > > > > | Deit-B | PGD | 20.4 | 18.6 | 28.9 | 19.4 | 24.8 | 35.2 | 44.8 |
> > > > > | Deit-B | AA | 14.4 | 13.6 | 16.5 | 12.4 | 22.4 | 21.6 | 32.8 |
> > > > >
> > > > > As shown in the table, PGD always achieves better attack success rate than AA, which implies that the adversarial transferability of AA can be worse than PGD.
> > > > >
> > > > > [g]. On Improving Adversarial Transferability of Vision Transformers. ICLR 2022 Spotlight.

---

> > > > > > ### Comment · Reviewer_ioBB · 2022-08-08
> > > > > > **Response**
> > > > > >
> > > > > > Dear authors,
> > > > > >
> > > > > > I still don't fully agree with the explanation.
> > > > > > Let's focus on one of the attacks, A-PGD, and leave aside the other 3 attacks.
> > > > > >
> > > > > > A-PGD should perform similar to PGD. What is your performance on A-PGD?
> > > > > >
> > > > > > Now there are two cases:
> > > > > >
> > > > > > 1) If the results are similar to PGD, this means that AA results should be at least similar to PGD.
> > > > > >
> > > > > > 2) If A-PGD and PGD results are not similar, it requires an explanation of the reason that adaptive learning rate gives a large gap.

---

> > > > > > > ### Author Response · Authors · 2022-08-09
> > > > > > > **Response to Reviewer ioBB (Cont.)**
> > > > > > >
> > > > > > > ### Re Auto-Attack Results (Cont.)
> > > > > > >
> > > > > > > Thanks for your reply. We first evaluate RNA under four attacks in Auto-Attack separately, including APGD, APGDT, FAB, and Square. All the attack settings are the same as those in Auto-Attack. The accuracy or RNA with ResNet-18 on CIFAR-10 under different attacks are reported in the following table:
> > > > > > >
> > > > > > > | PGD | APGD | APGDT | FAB | Square | AA |
> > > > > > > | ---- | ----- | ------ | --- | ------ | --- |
> > > > > > > | 60.69 | 63.31 | 72.12 | 81.56 | 78.84 | 65.61 |
> > > > > > >
> > > > > > > As shown in the table, there exists a gap between PGD and APGD/AA. We mainly attribute this gap to the efficient designing of APGD and Auto-Attack where the attacks are terminated when a successful adversarial example is found. Although it is effective and efficient in deterministic models, it might not work for randomized models, such as RNA.
> > > > > > >
> > > > > > > Some relevant work also discusses the performance of Auto-Attack against stochastic models. For example, [h] mentioned
> > > > > > > > “One of the reasons is that AutoAttack is optimized for efficiency and so each of its attacks is usually terminated once a misclassification occurs. This is applicable to deterministic models, but for stochastic ones such as an RT defense, the adversary is better off finding the adversarial examples that maximize the expected loss instead of ones that are misclassified once” and “AutoAttack is ineffective against randomized models”
> > > > > > >
> > > > > > > in Section 6.5, which corresponds to our explanation.
> > > > > > >
> > > > > > > Finally, we argue that the major contribution of our work lies in the defense via the normalization aggregation, which explores and utilizes the adversarial transferability among different normalizations. The reason why Auto-Attack is not effective in adversarial transferability is actually out of the scope. Even the current work in ICML 2022 [h] only provides a possible explanation of the poor performance of Auto-Attack against randomized models. Thus, it remains an open problem for further research. Another new work might be required to further analyze the reasons. We thank the reviewer for the in-depth understanding of Auto-Attack and comments.
> > > > > > >
> > > > > > > [h]. Demystifying the Adversarial Robustness of Random Transformation Defenses. ICML 2022.

---

> > > > > > > > ### Comment · Reviewer_ioBB · 2022-08-09
> > > > > > > > **Response**
> > > > > > > >
> > > > > > > > Dear authors,
> > > > > > > >
> > > > > > > > Thank you for your response.
> > > > > > > >
> > > > > > > > Regarding the results -- if A-PGD achieves 63.31, this number should be AA's upper bound, since A-PGD is the first to run. Why AA has 65.61? It doesn't adds up.
> > > > > > > >
> > > > > > > > regarding the setting of AA -- did you ran the random version (version='rand')? this will be the fair comparison.

---

> > > > > > > > > ### Author Response · Authors · 2022-08-09
> > > > > > > > > **Response to Reviewer ioBB (Cont.)**
> > > > > > > > >
> > > > > > > > > ### Re Auto-Attack Results (Cont.)
> > > > > > > > > Thanks for your reply again. We carefully checked the results and logs of APGD and found that the gap between APGD and AA in the previous table is a mistake$^{*}$. Sorry for the confusing results. We report the correct results of four attacks in Auto-Attacks separately in the following table:
> > > > > > > > >
> > > > > > > > > | Method | PGD | APGD | APGDT | FAB | Square | AA |
> > > > > > > > > | ------- | ---- | ----- | ------ | --- | ------ | --- |
> > > > > > > > > | RNA | 60.69 | 65.33 | 71.28 | 81.86 | 79.84 | 65.61 |
> > > > > > > > >
> > > > > > > > > As shown in the table, both AA and APGD achieves the similar results. The remaining small gap $0.28$ could be a result of the randomness of our algorithm. The reason why APGD has similar performance to AA mainly lies in the ratio of adversarial examples generated by APGD is much larger than the others. Taking the results of ResNet-18 on CIFAR-10 as an example, the ratio of adversarial examples generated by APGD is $0.7965$, the ratio of adversarial examples generated by other attacks is $0.0118\%$, and the ratio of remaining clean examples is $0.1917$. Thus, AA and APGD have the similar results.
> > > > > > > > >
> > > > > > > > >
> > > > > > > > > ### Re Setting of AA
> > > > > > > > > Following the setting in other randomized defense work [38], the version of AA is set to ‘standard’, which is also same to the official implementation of [g, h].
> > > > > > > > >
> > > > > > > > > As suggested, to further verify the performance of RNA in EOT format of AA, we set the version of AA to ‘rand’ for evaluation of RNA with ResNet-18 on CIFAR-10 where the gradients in $APGD_{ce}$ and $ APGD_{dlr}$ is computed by the expectation of $20$ EOT iterations. The results are reported in the following table:
> > > > > > > > >
> > > > > > > > > | Method | AA$_{standard}$ | AA$_{rand}$ |
> > > > > > > > > | ------- | --------------- | ---------- |
> > > > > > > > > | RNA | 65.61 | 63.38 |
> > > > > > > > >
> > > > > > > > > As shown in the table, the rand version of AA achieves slightly higher attack success rate. Compared with other normalization baselines, RNA still shows superiority. We mainly attribute it to the designed random space with lower transferability, which makes it difficult to compute appropriate gradient via a simple expectation.
> > > > > > > > >
> > > > > > > > > [g]. On Improving Adversarial Transferability of Vision Transformers. ICLR 2022 Spotlight.
> > > > > > > > >
> > > > > > > > > [h]. Demystifying the Adversarial Robustness of Random Transformation Defenses. ICML 2022.
> > > > > > > > >
> > > > > > > > > \* Most existing works use AA in their experiments, we also follow this tradition and use the standard code. This is our first time to conduct breakdown experiments of AA, we did a quick implementation during the rebuttal. Thanks to the thoughtful comments of the reviewer, we fixed a bug in this breakdown experiment. Obviously, this breakdown experiment and its code are isolated from the major experiment of the paper. Thus, there is no potential influence on the correctness of the whole paper.

---

> ### Author Response · Authors · 2022-08-07
> **Any further questions?**
>
> We appreciate your efforts in reviewing our paper. We have tried our best to provide the experimental results as suggested. Would you mind checking our response, and is there any further question we need to clarify?

---

### Official Review · Reviewer_q4wP · 2022-07-06

**Rating:** 7
**Confidence:** 5
**Soundness:** 3 good
**Presentation:** 3 good
**Contribution:** 3 good

**Summary:**

Inspired by the limited adversarial transferability across different normalizations, the authors proposed to involve randomness into the types of normalization layers and introduce a RNA module that reduces the adversarial transferability in a pre-defined random space, which improves the defense against adversarial examples. To evaluate the effectiveness of their algorithm, the authors provided experimental results on CIFAR-10/100 and ImageNet.


**Questions:**

As mentioned above, several questions are listed below:

1. What is the risk of random sampling strategy? What is the probability of sampling the similar paths?

2. How about the involvement of randomness in weight parameters and architectures? What is the advantage of proposed RNA compared with these baselines?

3. Is there empirical evidence that the smoothness of different normalization layers controls the defense performance?


**Limitations:**

The limitations have been addressed and there is no potential negative societal impact of this work.

**Strengths And Weaknesses:**

The authors studied a simple yet effective randomized mechanism with normalization layers. In general, the paper reads well, and the presentation is clear. The paper is original in that it studies the connections between normalization layers and adversarial defense. The theorical analysis well explains the principle of proposed algorithm. The paper includes a clear experimental setup and a meticulous comparison with other variant baselines as well as state-of-the-art algorithms on popular benchmarks. The results seem convincing.

Despite its contributions, I have several concerns:

1. The authors mentioned that they adopt random sampling strategy to utilize the adversarial transferability in random space. However, there exists a chance that the similar paths are sampled during both attack and inference stage, and there is no discussion of this scenario as well as the probability of it.

2. Besides normalization layers, there exist wide components in the networks which can involve randomness, such as weight parameters and architectures. Although the transferability evaluation in Fig 1 shows that there exists a poor transferability of adversarial examples among different types of normalization layers, it seems to me that this transferability also holds true for weight parameters and architectures. There is no discussion of these variant baselines.

3. The authors claimed that the smoothness of different normalization layers directly controls the adversarial transferability. However, it is hard for me to find corresponding evidence in experiment section.

---

> ### Author Response · Authors · 2022-08-02
> **Response to Reviewer q4wP**
>
> We thank the reviewer for the constructive comments and valuable suggestions.
>
> ###  Re Sampling probability.
>
> We take ResNet-18 as an example, which has 20 normalization layers. The probability of sampling the same path in both attacking and inference stage is $9.53 \times 10^{-5}$% while the probability of sampling paths with path difference of 10 is $17.62$%. Furthermore, the discussion of the influence of path difference has been included in Figure 5 (c) and Section 4.3. The stability of RNA has been also evaluated in Section 3 in our supplementary material.
>
> ###  Re Other random selection methods.
>
> The involvement of randomness in other random selection scenarios is also feasible for defense. However, our proposed RNA has several advantages. To begin with, compared with the selecting sub-network in parameters, RNA does not need to maintain an actual supernet where each path contains the similar capacity of popular architectures, such as ResNet and WideResNet. For example, in the field of Neural Architecture Search (NAS), it is always expensive and time-communing to train a supernet. Furthermore, we involve the randomness in the normalizations based on both the theoretical analysis in Section 2.3 and empirical evidence in Figure 3. According to the Theorem 2.1, the upper bound of adversarial transferability is controlled by the smoothness and gradient similarity, which all have strong connections to the normalization layers in our analysis, and we provide experimental results in Table 4. Moreover, RNA is also orthonormal to other random selection methods. With the involvement of different kinds of random selections, the random space could have potential lower adversarial transferability.
>
> ###  Re Empirical evidence that the smoothness of different normalization layers controls the defense performance.
>
> In Figure 2, we visualize the smoothness of different normalization in both 3D and 2D and find that IN is smoother than other normalization layers. In Table 4, we have tried several combinations of different normalization layers to form the random space. Comparing the performance of LN+BN and IN +BN, the smoother normalization layer IN achieves much better results than LN, which empirically verifies our theoretical analysis in Section 2.3.

---

> > ### Comment · Reviewer_q4wP · 2022-08-08
> > **replies to the authors' responses.**
> >
> > Dear authors,
> >
> > Many thanks for responding to my concerns on the "probability of sampling", "advantages comparisons," and "empirical evidence." My concerns have been fully resolved.
> >
> > I have read other reviews and replies, and there are no other issues from my end.
> > I would like to increase my rating score by one.
> >
> > Best,\
> > Reviewer q4wP

---

### Official Review · Reviewer_J8qn · 2022-07-08

**Rating:** 7
**Confidence:** 4
**Soundness:** 3 good
**Presentation:** 3 good
**Contribution:** 3 good

**Summary:**

The paper discusses the relation between normalization layers used in DNNs and adversarial transferability. The author(s) show(s) that the choice of normalization layer highly influences the success rate of transferred adversarial examples. In detail, it is shown that adversarial examples transfer worse (i.e., have a lower success rate of fooling the network) if a different normalization layer is used in the attacked network. This fact is used to motivate a novel technique to robustify existing neural network architectures: the key idea is to randomly select the used normalization layer during inference. Experiments on CIFAR-10, CIFAR-100 and ImageNet show that this approach is effective for the commonly used ResNet-18 and WideResNet32.

**Questions:**

What is meant with $\beta$ in Equation (7)?

**Limitations:**

Limitations and potential negative societal impact have not been addressed.

**Strengths And Weaknesses:**

STRENGTHS:
- The paper proposes an effective, clear and easy-to-implement method that is well-motivated by experimental and theoretical results.
- The extensive experimental evaluation on ResNet-18 and WideResNet32 architectures using a plethora of attacks (FGSM, PGD-20, CW, MIFGSM, DeepFool, AutoAttack) is convincing. An ablation study is conducted to show the effectiveness of the different components and the importance of different combinations of normalization layers.
- The paper presentation is well done and only contains only a few errors (see minor remarks). Used computational resources are mentioned.

WEAKNESSES:
- The used notation can be improved at some places:
    1. Equation (1): $x\sim\mathcal{X}, y\sim\mathcal{Y}$ suggests that $x$ and $y$ are drawn independently form the dataset. Here, $x, y\sim\mathcal{X},\mathcal{Y}$ would be correct. Also in Algorithm 1 $\{\mathcal{X},\mathcal{Y}\}$ has to be a tuple instead of set.
    2. In Equation (3): $\hat{y}(bgn)\_i^{(k)}$ suggests that $\hat{y}$ is a function in terms of $(bgn)$, which is not the case. I would suggest to use $\left(\hat{y}_{(\text{BGN})}^{(k)}\right)_i$ instead.
    3. Different symbols are used to denote multiplication (Equation (3) vs Equation (8))
- The stated Theorem 2.1 is hard to understand. Although, I see some value in the theorem, I think the author(s) should spend some time to reformulate it:
    1. It stated that "the gradient norm and $\beta$ in Equation (7) can be bounded as [...]", however, there is no $\beta$ in Equation (7) (only  $\beta_a$ and  $\beta_b$).
    2. The variable $T$ is introduced multiple times in a single sentence: "the level of adversarial transferability" and "attack success rate".

MINOR REMARKS:
- Typo "Adversarial Transfersability" in Abstract, "hessian" is sometimes not capitalized
- subscript and supperscript "gn" and "bgn" should not be typeset in math mode
- line 96: extra period in front of references [5, 20]

---

> ### Author Response · Authors · 2022-08-02
> **Response to Reviewer J8qn**
>
> We thank the reviewer for the constructive comments and valuable suggestions.
>
> ###  Re Improve notation.
>
> We will improve all the notations in the revised manuscript. Thanks for your suggestions.
>
> ###  Re $\beta$ in Equation 7.
>
> The $\beta_a$ and $\beta_b$ in Eq. 7 denote the upper bounds of gradient norm of network $h_a$ and $h_b$ which correspond to two paths randomly sampled from RNA. For simplicity, we dismiss the usage of subscript in the analysis of $\beta$ in Eq. 8.

---

### Official Review · Reviewer_kqKQ · 2022-07-11

**Rating:** 6
**Confidence:** 3
**Soundness:** 3 good
**Presentation:** 4 excellent
**Contribution:** 3 good

**Summary:**

This work provides a clear explanation for the relationship between normalizations and adversarial transferability, which further inspires the authors to propose a random normalization aggregation method where the adversarial robustness can be significantly improved.

**Questions:**

I suggest the authors give an explanation for the difference between RNA (random selection in normalizations) and other random selection methods, i.e., selecting sub-network in parameter or feature space.

**Limitations:**

Please see [Strengths And Weaknesses]

**Strengths And Weaknesses:**

Strengths:

1. Using adversarial transferability to boost adversarial robustness is interesting and inspiring, which is technical and methodology sound.

2. Establishing the relationship between normalizations and adversarial transferability is technically sound, and, in my opinion, it will make some impact on our community and give some insights to more researchers.

Weaknesses:

1. The connection between the mentioned novel viewpoint of using adversarial transferability and improved adversarial robustness should be highlighted, i.e., why transferability can contribute to boosting robustness, similarly why RNA can contribute to boosting robustness, moreover, do these two reasons share the same inspiration?

2. For the introduction to motivation, the authors claim that (line47-line50) the performance gap between BN and other normalizations results from “the adversarial transferability among normalizations,” which is a bit confusing and needs further justification. Specifically, the diagonal results are evaluated using white-box attacks, and the robust accuracy is about 50%, so that the gap may come from the difference between white- and black-box attacks. As this is an essential part of inspiring the authors to investigate the relationship between transferability and normalizations (line50), I suggest the authors provide a detailed justification.


3. One concern is that the robustness may come from the designed random selection operation, so it is necessary to employ expectation over transformation, EOT. [1], for the robustness evaluation to defend against the random operation.

Typos:

\tilde{x} makes Eq.2 confusing.

[1] Synthesizing Robust Adversarial Examples

---

> ### Author Response · Authors · 2022-08-02
> **Response to Reviewer kqKQ**
>
> We thank the reviewer for the constructive comments and valuable suggestions.
>
> ###  Re The connection between adversarial transferability and adversarial robustness.
>
> RNA builds a "supernet" which contains exponential number of paths with different normalization layers. Since we apply random sampling strategy during inference stage, the generated adversarial examples of sampled attacking path under white-box attacks are always fed to another different path for inference. This scenario corresponds to the transfer-based black-box attacks. Thus, the defense performance of this "supernet" built by RNA is actually determined by the level of adversarial transferability of this random space. In other words, the adversarial robustness can be boosted if the adversarial examples of paths in this "supernet" are difficult to be transferred to other paths. That is why adversarial transferability and adversarial robustness are strongly connected in our setting.
>
> The reason why RNA can contribute to adversarial robustness mainly lies in the three components in Section 3, including path increment in random space, black-box adversarial training, and normalization types selection. We explain the contribution of them as follows:
> * Path increment in random space: One of the key factors of this defense mechanism lies in the huge number of possible paths, which guarantees that attacking path and inference path are different. To achieve this objective, RNA randomly samples normalization layers for each layer in the network, which exponentially increases the number of paths.
> * Black-box adversarial training: One of the most effective approach to improve adversarial robustness is adversarial training. We incorporate RNA into adversarial training in a straightforward manner to form a "black-box" adversarial training algorithm. Specifically, we randomly sample paths for both adversarial example generation and network optimization, which directly reduces the adversarial transferability among paths, and thus improves the adversarial robustness.
> * Normalization types selection: We provide both theoretical analysis of the connections between adversarial transferability and the type of normalization layers in Section 2.3 as well as the discussion of gradient similarity with empirical evidence in Figure 3. Specifically, the normalization layers with smaller group size tend to have lower adversarial transferability, and the normalization layers in different categories tend to have lower adversarial transferability. Thus, RNA selects BN and IN to form the random space, which reduces the overall adversarial transferability among possible paths.
>
>
> ###  Re The performance gap between BN and other normalizations (Line 47-50).
>
> In Line 47-50, we discuss the gap between BN under white-box attacks and other normalization under black-box attacks. The gap we discuss in the paper refers to the gap between accuracy under white-box attacks (around 50%) and the one under black-box attacks (around 70%). Transfer-based black-box attacks naturally achieve lower attack success rate than white-box attacks, which motivates us to involve black-box setting in adversarial defense. According to the results of Figure 1 (c) and (d), we find a large gap between white and black-box accuracy. We attribute this gap to the adversarial transferability among normalization layers, which motivates us to explore the connections between normalization layers and adversarial transferability. We will revise this confusing claim in the revised manuscript.
>
> ###  Re Expectation over transformation.
>
> We apply the EOTPGD attack from TorchAttacks [26] to evaluate the performance of RNA. The results are shown in the following table.
>
> | Network | dataset | EOT PGD$^{20}$ | PGD$^{20}$ |
> | ------- | ------ | --------- | --------- |
> | ResNet-18 | C-10 | 59.71 | 59.79 |
> | ResNet-18 | C-100 | 34.85 | 35.51 |
> | WideResNet | C-10 | 62.04 | 60.49 |
> | WideResNet | C-100 | 36.65 | 35.98 |
>
> We compare EOTPGD and PGD with ResNet-18 and WideResNet on both CIFAR-10 and CIFAR-100. RNA achieves the similar performance under both EOTPGD and PGD.

---

> > ### Author Response · Authors · 2022-08-02
> > **Response to Reviewer kqKQ (Cont.)**
> >
> > ###  Re Other random selection methods.
> >
> > The involvement of randomness in other random selection scenarios is also feasible for defense. However, our proposed RNA has several advantages. To begin with, compared with the selecting sub-network in parameters, RNA does not need to maintain an actual supernet where each path contains the similar capacity of popular architectures, such as ResNet and WideResNet. For example, in the field of Neural Architecture Search (NAS), it is always expensive and time-communing to train a supernet. Furthermore, we involve the randomness in the normalizations based on both the theoretical analysis in Section 2.3 and empirical evidence in Figure 3. According to the Theorem 2.1, the upper bound of adversarial transferability is controlled by the smoothness and gradient similarity, which all have strong connections to the normalization layers in our analysis, and we provide experimental results in Table 4. Moreover, RNA is also orthonormal to other random selection methods. With the involvement of different kinds of random selections, the random space could have potential lower adversarial transferability.

---

> > ### Comment · Reviewer_kqKQ · 2022-08-07
> > **Response to rebuttal**
> >
> > Dear authors,
> >
> > My concerns/questions are partially addressed. The setting used in ``Re Expectation over transformation'' is a bit confusing. Specifically, EOT introduces more random transformations for generating adversarial examples, but the authors use the same PGD iteration. I suggest reporting results with more iterations, e.g., more iterations under each transformation and more transformations.
> >
> > If the author can address the concern, I will raise my rating.

---

> > > ### Author Response · Authors · 2022-08-07
> > > **Re Expectation over transformation (Cont.)**
> > >
> > > ### Re Expectation over transformation (Cont.)
> > >
> > > Thanks for your reply. We clarify the setting in EOTPGD. EOTPGD aims at attacking randomized model. Following TorchAttacks [26] and official implementation of [b], EOT is always combined with PGD/FGSM attack to evaluate the defense performance of randomized model, which corresponds to our setting. Thus, we conduct experiments to evaluate RNA under EOT+PGD (EOTPGD) attack using TorchAttacks [26].
> > >
> > > In [a], Expectation over Transformation (EOT) is suggested to compute the gradient over the expected transformation to the input. In EOTPGD, instead of computing the gradient of one sampled path in each PGD iteration, the gradient is computed based on the expectation of $N$ sampling times due to the randomness of network, which we strictly follow the attack setting in [26, b]. As suggested in the comments, we increase $N$ to $100$ to evaluate the performance of RNA with more EOT iterations. Specifically, the attack iteration of PGD is set to $20$ and **in each PGD iteration**, the iteration of EOT is set to $100$. The results are reported in the following table:
> > >
> > >
> > > | Network | dataset | EOT$^{100}$ PGD$^{20}$ | PGD$^{20}$ |
> > > | ------- | ------ | --------- | --------- |
> > > | ResNet-18 | C-10 | 65.27 | 59.79 |
> > > | ResNet-18 | C-100 | 38.84 | 35.51 |
> > >
> > > As shown in the table, the attack success rate becomes lower if EOT with more iterations is applied. We mainly attribute it the designing of random space of RNA. Due to the lower adversarial transferability among the paths as well as the exponential number of paths in this random space, it is difficult for EOT attacks to compute the accurate gradient with expectation.
> > >
> > > [a]. Synthesizing Robust Adversarial Examples. ICML 2018.
> > >
> > > [b]. Learn2Perturb: an End-to-end Feature Perturbation Learning to Improve Adversarial Robustness. CVPR 2020.

---

> > > > ### Comment · Reviewer_kqKQ · 2022-08-07
> > > > **Thanks for your prompt reply.**
> > > >
> > > > Dear authors,
> > > >
> > > > Thanks for your prompt reply, the experiments addressed my concerns, and so I raise my rating.

---

### Author Response · Authors · 2022-08-09
**Thank All Reviewers**

We sincerely appreciate all the reviewers for their valuable comments and suggestions. We have revised the manuscript according to the comments. The updated contents are highlighted by blue text in the revised manuscript.

---

### Meta-Review · Area_Chair_tf7s · 2022-08-26

**Recommendation:** Accept
**Confidence:** Less certain

**Metareview:**

This paper intrdouces the relation between normalizations and adversarial transferability, and proposes a method using random normalization aggregation for enhancing adversarial robustness.

Three reviewers agreed with the interesing idea, thorough expreiments, theroetical analysis, and the effectivess, so they gave acceptance score.

However, one reviewer (ioBB) raised a concern on the results of Auto-attack (AA) and lack of in-depth discussion on the results.

Unfortunately, AC and the reviewers failed to make a consensus on decision during the discussion period, .

AC carefully read the paper, the rebuttal, and the reviewers' discussion. The main remaining issue raised by Reviewer-ioBB is it is not clear the reason why the results of AA are highter than PGD. The authors provided more extensive experimental results, focusing on comparing the results of AA (and breakdown of four AA) and PGD. Also, they conjecture these result from handling adversarial transferability under randomness-based method via normalization aggregation.

AC also agrees with the authors and Reviewer-q4wp that the in-depth analysis on the reason of AA-PGD results is ouf-of-scope of this paper and theses results are consistent to recent works on adversarial transferability. So, this issue might be left as future work.

Because it seems that the contribution of this paper is enough for machine leanring community except the discussion on the AA-PGD reason, AC recommends accepting this paper.



**Award:**

No

---

### Decision · Program_Chairs · 2022-09-14

Accept